# Pulsatile inputs achieve tunable attenuation of gene expression variability and graded multi-gene regulation

Dirk Benzinger[1] & Mustafa Khammash[1]

Many natural transcription factors are regulated in a pulsatile fashion, but it remains unknown whether synthetic gene expression systems can benefit from such dynamic regulation. Here we find, using a fast-acting, optogenetic transcription factor in *Saccharomyces cerevisiae*, that dynamic pulsatile signals reduce cell-to-cell variability in gene expression. We then show that by encoding such signals into a single input, expression mean and variability can be independently tuned. Further, we construct a light-responsive promoter library and demonstrate how pulsatile signaling also enables graded multi-gene regulation at fixed expression ratios, despite differences in promoter dose-response characteristics. Pulsatile regulation can thus lead to beneficial functional behaviors in synthetic biological systems, which previously required laborious optimization of genetic parts or the construction of synthetic gene networks.

[1] Department of Biosystems Science and Engineering (D-BSSE), ETH–Zürich, Mattenstrasse 26, Basel 4058, Switzerland. Correspondence and requests for materials should be addressed to M.K. (email: mustafa.khammash@bsse.ethz.ch)

The relationship between gene expression and cellular phenotype lies at the center of many questions in different branches of biological research. While strong perturbations of gene expression like knock-outs and overexpression led to a tremendous increase in our understanding of protein function, graded gene expression regulation allows us to obtain a quantitative understanding of the expression–phenotype mapping. Furthermore, conditional and titratable gene expression is of major importance in biotechnology and synthetic biology. Thus, a variety of tools for regulating cellular protein levels, such as gene expression systems based on hormone or light-inducible transcription factors, were developed[1]. With a few exceptions[2–4], expression levels are regulated by adjusting the strength of an input, leading to a graded and constant activation of a transcriptional regulator (Fig. 1a, from here on referred to as amplitude modulation (AM)). In contrast, recent studies have shown that many natural regulatory proteins, including transcription factors (TFs), exhibit pulsatile activation patterns leading to a variety of phenotypic consequences[5].

Motivated by the occurrence of pulsatile transcription factor regulation in natural systems, we hypothesized that synthetic gene expression systems can benefit from such dynamic regulation. To test this hypothesis, we constructed a fast-acting, and genomically integrated, optogenetic gene expression system based on the bacterial light-oxygen-voltage protein EL222 in *Saccharomyces*

*cerevisiae*[4]. Fast kinetics of the optogenetic TF together with the ability to control light intensity with high-temporal precision allowed us to tune gene expression using pulsatile TF inputs. In particular, we performed pulse-width modulation (PWM)[3], meaning that the duration of input pulses is varied to achieve different gene expression levels, while keeping the period of the pulses constant (Fig. 1b). The ratio of pulse duration to the period is referred to as duty cycle. PWM can be performed at different input amplitudes and periods, providing further options for dynamic signal encoding to regulate gene expression levels. We used a mathematical model to identify suitable PWM periods and then showed experimentally that these can be exploited to tune gene expression properties. By comparing this PWM approach to AM, we establish that dynamic encoding of pulsatile signals can drastically increase the functionality of gene expression systems.

## Results

**Characterizing an EL222-based expression system in yeast.** In order to regulate gene expression using PWM, we implemented an optogenetic gene expression system based on a previously described TF consisting of a nuclear localization signal, the VP16 activation domain[6], and the light-oxygen-voltage domain protein EL222 of *Erythrobacter litoralis* (VP-EL222)[4]. Blue-light illumination triggers structural changes in EL222 leading to homodimerization and binding to its cognate binding site (Fig. 1c). An

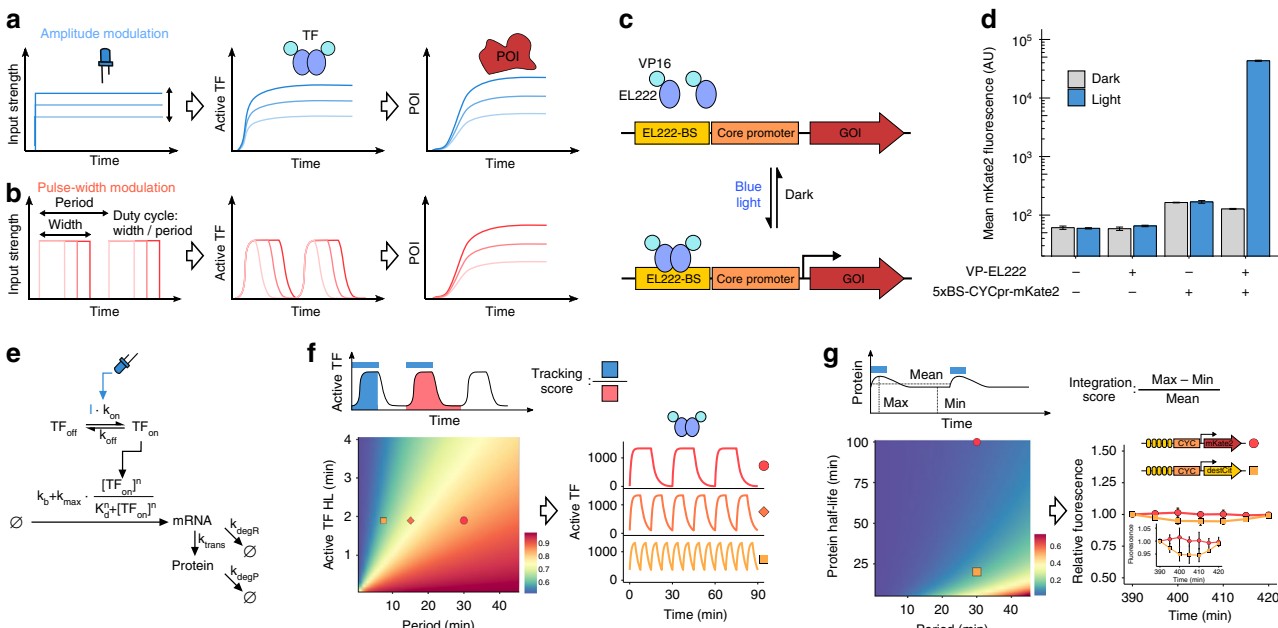

**Fig. 1** Characterization of an EL222-based optogenetic gene expression system in *S. cerevisiae*. **a** and **b** Schematic of gene expression regulation by AM (**a**) and PWM (**b**). Input signals (left) lead to TF activation (middle) and expression of a protein of interest (POI, right). **c** Illustration of the optogenetic gene expression system. Blue-light triggers VP-EL222 dimerization and transcription of a gene of interest (GOI). **d** Effect of blue-light illumination on VP-EL222 mediated gene expression. Strains, with or without the VP-EL222 and a reporter construct (5xBS-*CYC180*pr-mKate2), were grown either in the dark or under illumination (460 nm, 350 μW cm$^{-2}$) for 6 h. Data represent mean and s.d. of three independent experiments. **e** Graphical representation of the model describing VP-EL222 mediated gene expression. I represents the light input. See Methods and Supplementary Note 1 for details. **f** Model-based analysis of pulsatile TF behavior. To quantify the temporal TF response, we use a tracking score defined by the ratio between the integrated TF activity during the light pulse and the whole period (top, Supplementary Note 2). This metric is 1 if the TF activity perfectly tracks the input, and equals the duty cycle if TF activity does not change over time. The heatmap depicts the tracking score as a function of the PWM period and the half-life (HL) of the active VP-EL222 state (50% duty cycle, 420 μW cm$^{-2}$). On the right, predicted temporal TF activities are shown for PWM conditions marked on the heatmap after an initial settling period (see Supplementary Note 2). **g** Model-based analysis of PWM-mediated protein expression. The heatmap depicts the integration score (top), quantifying temporal variations of protein levels (Supplementary Note 2), as a function of the PWM period and protein half-life (10% duty cycle). On the right, the measured time course of FP expression in response to PWM with a 30 min period are shown after 390 min of induction at 10% duty cycle. Experiments were performed using mKate2 (red) and a destabilized mCitrine variant[8,9] (yellow). Fluorescence is normalized by the value measured after 390 min. Inset differs in *y*-axis scaling. Data represent the mean and s.d. of two independent experiments

EL222-responsive promoter was constructed by inserting five binding sites for EL222[4] upstream of a truncated CYC1 promoter (5xBS-CYC180pr) and was used to drive the expression of the fluorescent protein (FP) mKate2[7]. For initial characterization, we measured the expression levels of mKate2 in the dark and after 6 h of blue-light illumination via flow cytometry (Fig. 1d). Illumination led to a VP-EL222 dependent increase in cellular fluorescence of more than 250-fold. In the dark, the presence of VP-EL222 did not affect gene expression. Neither blue-light illumination nor VP-EL222 activation affected cell growth or constitutive gene expression under the experimental condition and timescales (Supplementary Fig. 1).

In order to achieve a quantitative understanding of the system and investigate potential PWM regimes, we derived a simple mathematical model of VP-EL222 mediated gene expression (Fig. 1e, for details see Supplementary Note 1). The model was fitted to the data of three characterization experiments, namely a gene expression time course, as well as dose response curves to AM and PWM with a period of 7.5 min (Supplementary Fig. 2). Analyzing the model showed the importance of fast VP-EL222 deactivation kinetics for successful PWM (Supplementary Note 2). For a fixed duty cycle, slow deactivation rates require long PWM periods to achieve purely pulsatile TF regulation, which is the desired regime of operation (Fig. 1f). However, such periods may result in significant temporal variation of downstream gene expression/input tracking (Fig. 1g). Here, the half-life of the active VP-EL222 state was inferred to be lower than 2 min (Fig. 1f, Supplementary Table 4). Measurements of transcription upon a blue-light pulse using single molecule fluorescent in situ hybridization (smFISH) lead to results consistent with the fast VP-EL222 kinetics (Supplementary Fig. 3, Supplementary Note 1). For the inferred deactivation rate, the model predicts strongly pulsatile TF activity for a 30 min PWM period and a 50% duty cycle, whereas for the same duty cycle TF activity does not return to baseline when a 7.5 min PWM period is used (Fig. 1f). Importantly, even for a 30 min period, temporal changes in protein expression at steady state are expected to be minor for a wide range of protein half-lives (Fig. 1g, Supplementary Note 2). We confirmed experimentally that there is no measurable input tracking for a stable fluorescent protein and relatively little input tracking for a destabilized FP with a half-live of ≈30 min[8,9] (Fig. 1g, see Supplementary Fig. 4 for characterization experiments regarding the destabilized FP). Thus, the fast kinetics of the VP-EL222-based system together with its tight regulation, and apparent lack of toxicity, makes it an ideal gene expression tool for a variety of applications and enables the regulation of protein levels by PWM.

**Pulsatile inputs achieve coordinated multi-gene expression.** Given that most cellular phenotypes are a result of the coordinated regulation of many genes whose protein expression ratios can be of high importance for achieving these phenotypes[10], we explored the use of AM and PWM for achieving graded expression of multiple proteins, each at a different level, with a single gene expression system. Eukaryotic genes are usually monocistronic and thus, promoter libraries are typically used to adjust relative expression levels[11]. Hence, we built a set of light-responsive promoters differing in the promoter backbone and EL222 binding site number. The resulting promoters showed a wide range of maximal expression levels with promoters based on both the GAL1 and the SPO13 backbone exhibiting very low basal expression (Fig. 2a, Supplementary Fig. 5). However, when we analyzed the response of two promoters differing in the number of EL222 binding sites to AM, we found that they show different dose-response behaviors (Fig. 2b, see Supplementary Fig. 6 for

model fits to the CYC180 promoter with two binding sites). In contrast, PWM with a period of 30 min resulted in coordinated expression with an almost linear relationship between the duty cycle and the protein output (Fig. 2c). Thus, only PWM is compatible with the use of a simple promoter library for graded multi-gene expression at constant ratios (Fig. 2d). We observed the same behavior when both reporters were located in a single cell (Supplementary Fig. 8a). Furthermore, a similar behavior was observed when different promoter backbones were used to tune expression levels (Fig. 2e, Supplementary Fig. 8b–e). We additionally found that the use of shorter PWM periods resulted in intermediate levels of coordinated promoter regulation, allowing for input-mediated tuning of expression ratios (Fig. 2d). This behavior is qualitatively recapitulated by the simple gene expression model (Supplementary Fig. 8f), suggesting that it results from TF activity deviating from a purely pulsatile activity regime at low PWM periods (Fig. 1f, Supplementary Note 2). Another possible explanation for this behavior may be promoter-dependent differences in the expression response to short TF pulses, which have been observed for natural promoters (see also Supplementary Note 1 'Refitting of promoter-specific model parameters')[12]. We note that Elowitz and colleagues have shown that frequency modulation of TF activity can coordinate multi-gene expression in S. cerevisiae[13]. Thus, our work demonstrates how we can learn from natural systems to better regulate gene expression in synthetic systems using simple strategies[13]. We further note that linear dose-response curves are exclusive to PWM regulation but can also be achieved for constant inputs by the construction of negative feedback loops[14].

**Reducing and tuning expression variability.** While, we have so far only analyzed the average response of cells to input signals, gene expression can exhibit a substantial amount of heterogeneity[15]. For many applications, precise single cell regulation of gene expression is desirable[16]. However, the ability to tune variability while keeping the mean expression fixed may allow for the analysis of its phenotypic consequences[15,17,18]. To date, variability regulation was achieved by the construction of synthetic gene networks[14,19–22] —namely feedback loops[14,20] and cascades[19,21,22]— as well as the tuning of promoter features, such as TATA boxes[17,19]. Variability reduction via frequency modulation of TF activity was previously proposed based on theoretical results[23]. In contrast, experimental work in S. cerevisiae has shown that oscillatory dynamics of the TF Msn2 can result in increased expression variability in a promoter-dependent fashion[12].

For the synthetic gene expression system, PWM led to reduced cell-to-cell variability in protein levels compared to AM for the same mean expression (Fig. 3a). Furthermore, changing the PWM period enabled tuning of expression heterogeneity with a single input and no change in network architecture (Fig. 3a). In order to investigate the mechanism behind this variability reduction, we performed a dual reporter experiment (see Methods for details). This assay allows for the decomposition of expression variability stemming from stochastic events at the promoter level (intrinsic) and global differences between cells (extrinsic)[24,25]. We found that PWM reduces both extrinsic and intrinsic variability (Fig. 3b, c). However, for most expression levels, extrinsic variability is the dominant source of heterogeneity in the synthetic expression system. Given that TF variability is thought to be a major determinant of extrinsic variability[26], we hypothesized that PWM leads to lower gene expression heterogeneity by operating in a promoter-saturating regime, where transmission of TF variability to gene expression output is minimal (Fig. 3d).

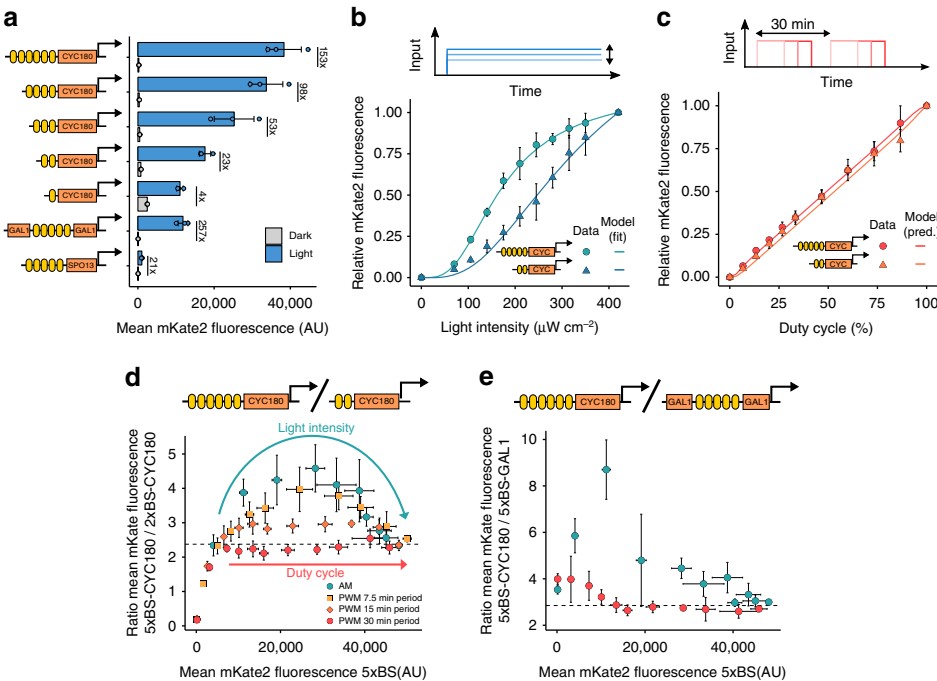

**Fig. 2** Coordinated multi-gene regulation using dynamic inputs. **a** A promoter library for gene expression at various expression levels. Schematics represent the different promoters. Yellow boxes represent EL222 binding sites and orange boxes represent partial sequences of yeast promoters. Strains, expressing mKate2 under the control of the respective promoter, were cultured for 6 h in the dark or the presence of blue light (350 μW cm⁻²). The average cellular mKate2 fluorescence was measured using flow cytometry. Data represent the mean and s.d. of three independent experiments. **b** and **c** Dose-response of two promoters to AM (**b**) and PWM (**c**). Strains expressing mKate2 under the control of either a *CYC180* promoter with five (circle, 5xBS) or two (triangle, 2xBS) VP-EL222 binding sites were grown under the illumination conditions depicted on the x-axis for 6 h. The light intensity and period for PWM were 420 μW cm⁻² and 30 min. Mean cellular fluorescence measurements were normalized to be 0 in the dark and 1 at the highest input level to allow for easy comparison. Non-normalized values are shown in Supplementary Fig. 7. Data represent the mean and s.d. of three independent experiments. Lines represent model fits or predictions. **d** Relative gene expression levels for different induction condition. Strains (as in **b** and **c**) were grown under the same illumination conditions (light intensity and duty cycle) as shown in **b** and **c**. In addition, the effect of the PWM period on coordinated expression was explored. The ratio of mKate2 expression from the 5xBS and the 2xBS promoter is plotted against the mKate2 expression from the 5xBS promoter for the same illumination conditions. The dashed line represents this ratio at constant illumination with a light intensity of 420 μW cm⁻². Data represent the mean and s.d. of three independent experiments. **e** Relative gene expression levels between a *CYC180* and *GAL1* based promoter with five binding sites for different induction condition. Experiments were performed as described in (**d**). Data represent the mean and s.d. of three independent experiments for the *CYC180*-based promoter and two independent experiments for the *GAL1*-based promoter

To approximate this phenomenon with our simple mathematical model, we performed simulations in which we drew TF concentration from a log-normal distribution describing the single-cell distribution of a mCitrine-tagged[27] version of VP-EL222 (Supplementary Note 3, Supplementary Fig. 11a). This model can qualitatively recapitulate the experimental data (Fig. 3e, Supplementary Fig. 11b, c). The model suggests that the ability to tune variability using the PWM period stems from active VP-EL222 concentrations residing at intermediate levels for extended time intervals at low PWM periods and short duty cycles (Fig. 3d, f, see also Supplementary Note 2). We further showed experimentally that PWM reduced the slope of the correlation between VP-EL222 expression levels and mKate2 output (Fig. 3g, Supplementary Fig. 11d). Next, we expressed VP-EL222 from a centromeric plasmid to increase TF variability by introducing plasmid copy number variation (Supplementary Fig. 12)[9]. Under these conditions, AM led to a wide-spread multi-modal protein distribution at intermediate expression levels (Fig. 3h). In contrast, PWM resulted in the merging of these distributions. Thus, the use of PWM does not only reduce heterogeneity as measured by the CV but may also lead to qualitatively different distributions by attenuating the effects of TF variability on downstream gene expression.

**A stochastic model recapitulates expression variability**. Using a simple ODE-based gene expression model we were able to show that the proposed mechanism of PWM-mediated noise reduction (by attenuation of upstream TF variability) stands in qualitative agreement with our experimental data. However, the model overestimated gene expression variability and did not consider intrinsic variability, which was significant at low expression levels (Fig. 3b, e, Supplementary Note 3). We thus sought to investigate whether a slightly more detailed, stochastic model can quantitatively recapitulate the experimentally observed expression variability (see Supplementary Note 4 for details regarding the modeling approach). To do so, we calibrated fluorescence values to cellular protein numbers using reference strains[28] (Supplementary Fig. 13a, Supplementary Note 4). As a dynamic source of VP-EL222 variability, we explicitly model its transcription and translation, meaning we model the extrinsic variability of the target gene as intrinsic variability of an upstream TF[29] (Fig. 4a, Supplementary Fig. 13b, and Supplementary Note 4). Prompted by our recent observation that VP-EL222 mediated transcription occurs in bursts whose timing as well as duration are affected by VP-EL222 activity[30], we modeled reporter gene transcription based on a two-state promoter model[25], with VP-EL222 activity positively regulating promoter activation (switching from a

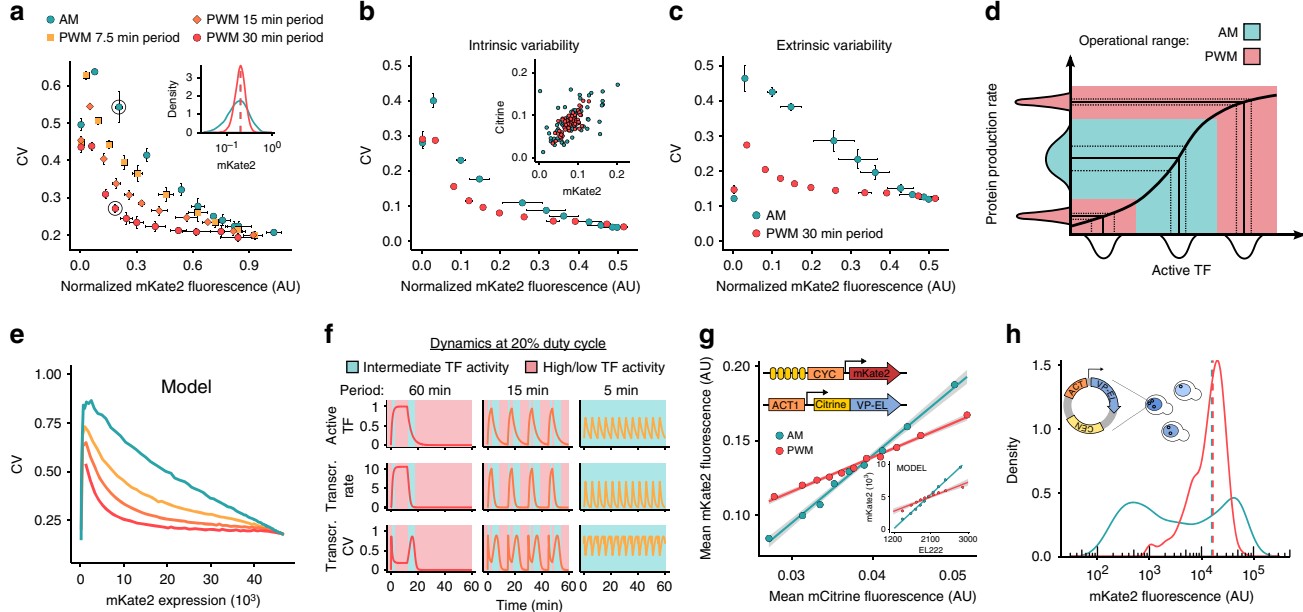

**Fig. 3** Effects of PWM and AM on gene expression variability. **a** Cell-to-cell variability, measured by the coefficient of variation (CV), as function of mean expression levels. Fluorescence was normalized by side-scatter measurements (Methods). Cells containing 5xBS-*CYC180*pr-mKate2 were induced for 6 h. Illumination conditions are identical to those in Fig. 2d. Data represent mean and s.d. of three independent experiments. The inset shows fluorescence distributions for the circled data points. **b** and **c** Contributions of intrinsic (**b**) and extrinsic (**c**) variability. Experiments were performed using a dual-color reporter strain (Methods)[24,25] under the same conditions as in (**a**). Data represent mean and s.d. of three independent experiments. The inset shows data of 60 cells for conditions with similar mean for AM (105 μW cm$^{-2}$) and PWM (13% duty cycle). **d** Schematic illustration of variability transmission from TF concentration to gene expression[57]. Distributions represent cell-to-cell variability. Assuming identical input–output relationships between cells, expression variability increases with the steepness of the input–output function. **e** CV as function of mean expression obtained using the 5xBS-*CYC180*pr model, including VP-EL222 variability. Induction regimes (line color) are as in (**a**). **f** Differences in TF dynamics affect transcriptional variability. Using the model, we analyzed how the PWM period effects temporal changes of active TF concentration (top, relative to maximal activity), transcription rate (middle, in mRNA min$^{-1}$), and transcription variability (bottom). Red shading represents intervals of high or low (>90% or <10% of maximum value) TF activity and blue shading represent intervals of intermediate TF activity. **g** Dependence of gene expression output on mCitrine-VP-EL222 levels. Fluorescence was analyzed after 1 h of induction. Cells were collected in 10 bins of equal cell number based on their normalized mCitrine fluorescence. Data points represent the mean normalized mKate2 and mCitrine fluorescence of cells from each bin. Lines represent linear regressions. Induction conditions: AM = 70 μW cm$^{-2}$, PWM = 350 μW cm$^{-2}$; 30 min period; 26.7% duty cycle. Inset: Model results for equivalent conditions. **h** Effect of AM (blue) and PWM (red) on fluorescence distributions with VP-EL222 expressed from a centromeric plasmid. Induction conditions: AM = 105 μW cm$^{-2}$, PWM = 420 μW cm$^{-2}$; 45 min period; 26.7% duty cycle. CV-mean relationship is shown in Supplementary Fig. 12b

repressed to a transcription competent state) and negatively regulating promoter deactivation (Fig. 4a, Supplementary Note 4).

By simulating the dual-color reporter experiment we found that this model can reproduce both intrinsic and extrinsic variability (Fig. 4b–d). Importantly, extrinsic variability is predicted based on characteristics of VP-EL222/the upstream TF (in particular protein copy number, variability, target dose-response curve, and degradation rate, see Supplementary Note 4) without additional parameter tuning, further increasing the support of the proposed mechanism of PWM-mediated variability reduction. Interestingly, the intrinsic variability behavior could not be reproduced by a model in which VP-EL222 directly affects the transcription rate (Supplementary Fig. 13c, d), suggesting that pulsatile TF activity may be capable of reducing variability resulting from transcriptional bursting[23]. We note, however, that future studies using more appropriate readouts, such as live-cell measurements of transcription[30,31], are required to study this aspect in detail.

We additionally found that the model nicely predicted the expression variability under different PWM periods (Fig. 4e), PWM amplitudes (Supplementary Fig. 13e), and for different

reporter protein degradation rates (Supplementary Fig. 13e), which may open the way to a model-predictive regulation of expression mean and variability.

**Mapping enzyme expression levels and variability to growth.** For most genes, it remains largely unknown to which degree variability in their expression affects phenotypic behavior. This lack of understanding partly stems from difficulties in tuning gene expression variability for the same mean level[18], despite recent progress in this area[22]. Using the properties of AM and PWM regulation, we sought to analyze how variability in the expression of the metabolic enzyme Ura3, which is required for survival of *S. cerevisiae* in uracil-free media[32], affects cellular growth at different expression levels. We found that mCitrine-tagged Ura3p is stable and that both AM and PWM-mediated expression does not affect cell growth in uracil rich media (Supplementary Fig. 14a, b). Thus, differences in dose-response curves between AM and PWM can be expected to stem from differences in variability (Supplementary Fig. 14c).

We found that, upon a shift from uracil-rich to uracil-free medium, the dose response of mean expression to growth depends on expression with tight regulation resulting in maximal

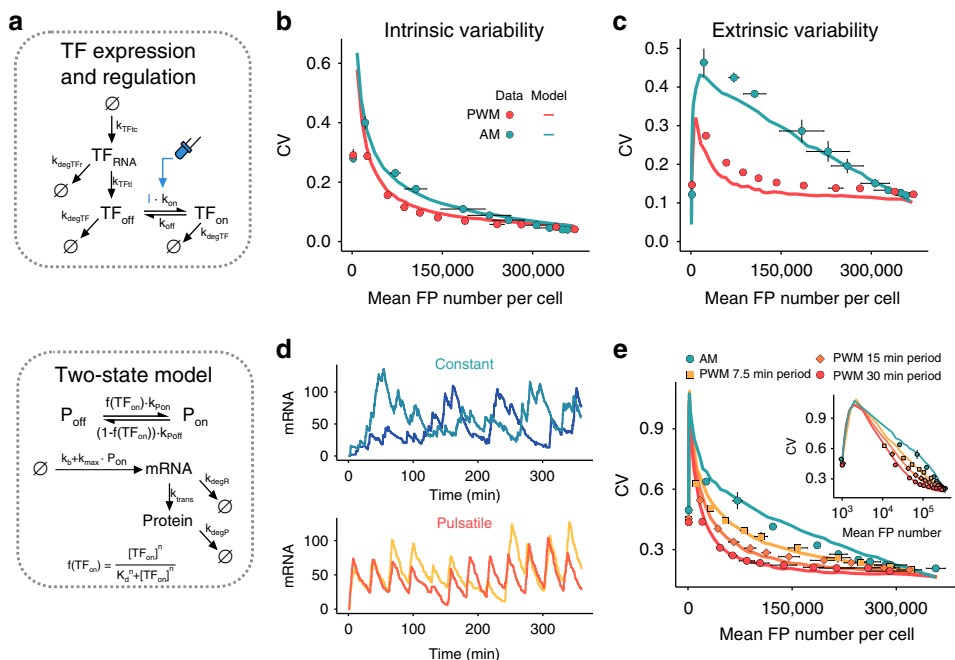

**Fig. 4** A stochastic model of VP-EL222 mediated gene expression. **a** Graphical representation of the stochastic model describing VP-EL222/TF expression and activation (top) and VP-EL222/TF mediated reporter gene expression (bottom). The light input is denoted by I. Arrows depict reactions. Details on the modeling approach can be found in Supplementary Note 4. **b** and **c** Modeling results (lines) and experimental data (circles) showing intrinsic (**b**) and extrinsic (**c**) contributions to gene expression variability (see Supplementary Table 6 for model parameters). A dual-color reporter experiment was simulated for 6 h and the variability decomposition procedure was applied to the simulated data as described in Methods section for the experimental data. Experimental results are the same as shown in Fig. 3b, c. **d** Intrinsic mRNA expression dynamics under constant (top, simulated intensity of 140 μW cm$^{-2}$) and pulsatile regulation (bottom, simulated PWM period of 30 min with 20% duty cycle and 420 μW cm$^{-2}$ light intensity). Simulations were performed without the incorporation of extrinsic variability, meaning with fixed levels of VP-EL222 expression (24,600 proteins per cell). Two single-cell traces are shown per condition. **e** Model predictions for the single 5xBS-*CYC180*pr reporter strain under different AM and PWM conditions. Data are the same as shown in Fig. 3a (see Supplementary Table 6 for model parameters)

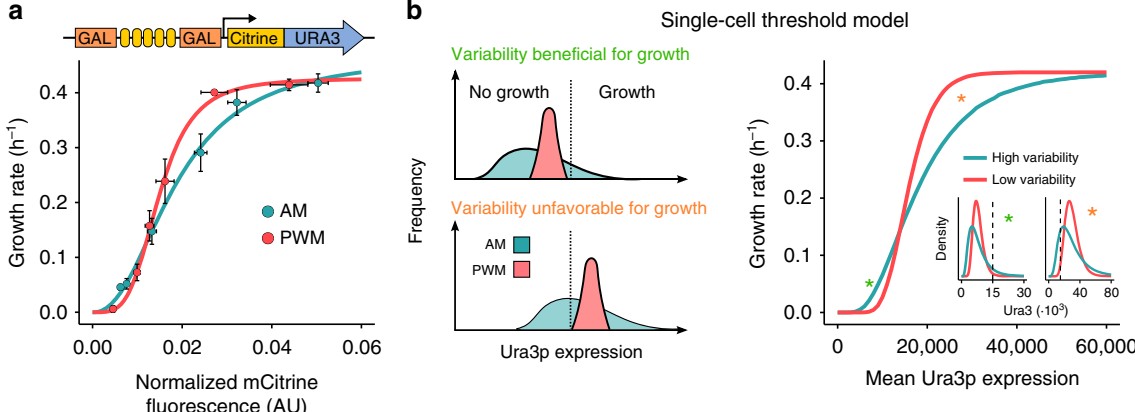

**Fig. 5** Effect of regulating URA3 expression levels by AM and PWM on cell growth. **a** Cells expressing mCitrine-tagged Ura3p from the 5xBS-*GAL1*pr were grown under AM (blue) and PWM (red) light induction for 14 h in media with uracil before transfer to uracil-free media, fluorescence, and growth measurements. PWM was performed with a 30 min period and 420 μW cm$^{-2}$ light intensity. Growth was measured for 5 h. Hill functions were fit to the data for comparison and guidance (see Supplementary Fig. 14f and Supplementary Table 7 for parameters). The effect of AM and PWM on the CV is shown in Supplementary Fig. 14c. Data represent the mean and s.e.m of three independent experiments. Data of all individual experiments are shown in Supplementary Fig. 14d, e. **b** The experimental data are consistent with an all-or-none growth response to Ura3p expression on the single-cell level (threshold model). A schematic representation illustrating how cell-to-cell variability combined with a threshold expression level required for growth (dashed line) may affect cellular growth for two different expression levels is shown on the left. For low mean expression levels, expression variability enables growth of a small subpopulation. In contrast, for high expression levels, cell-to-cell variability may lead to a subpopulation that is unable to grow and a subpopulation with unnecessarily high expression levels. The expected relation between mean expression levels and growth rate for the threshold model is shown on the right. Growth rates were calculated by computing the fraction of log-normal distributions with values higher than a threshold level (15,000) and multiplying the resulting fraction by the maximal growth rate of 0.42 h$^{-1}$. The CV of the log-normal distributions used was 0.6 for the high variability curve (blue) and 0.3 for the low variability curve (red). The insets show the distributions for mean expression levels marked by asterisks in the main figure together with the growth threshold value (dashed line)

growth with less enzyme expression (Fig. 5a, Supplementary Fig. 14d–f). The data further suggest that variability can result in increased growth at very low mean expression levels. We thus show that by comparing AM to PWM regulation, the influence of expression variability on cellular phenotypes can be investigated. In addition to its importance in basic research[18], such analysis may be relevant for the adjustment of optimal protein levels for synthetic biology applications[33] in which metabolic burden is non-negligible[34]. Furthermore, our results indicated that expression variability in a cell population may disguise the steepness of dose–response curves on the single-cell level (Supplementary Fig. 14f)[35], thus exemplifying the importance of precise regulation for the analysis of expression–phenotype relationships. To this effect, we found that the observed behavior is consistent with a simple model, in which cellular growth depends on *URA3* expression in an all-or-none fashion (Fig. 5b).

## Discussion

We present a highly inducible, fast-acting optogenetic expression system for *S. cerevisiae* which enables the regulation of protein levels by PWM. Learning from the use of pulsatile regulation in a natural system[13], we show that PWM enables the use of simple promoter libraries and a single input for the graded and coordinated regulation of multiple genes at different expression levels. We further propose a novel method for variability reduction and tuning in gene expression systems based on pulsatile inputs. Thus, the simple use of dynamic inputs may replace laborious optimization of promoter dose–response curves[36] and construction of gene networks[14,20,22] for synthetic biology and basic research applications[1,18]. Recently, a similar VP-EL222-based optogenetic expression tool was employed to regulate metabolic fluxes in a biotechnological setting[37] and it will be interesting to see whether the properties of PWM regulation described here can be beneficial for bioprocesses.

Our combined experimental and modeling work identified the attenuation of upstream TF variability as a mechanism behind PWM-mediated expression noise reduction and tuning. Not only could this mechanism be relevant for the control of single genes but also for the precise regulation of gene expression networks[38]. We further revealed a reduction of intrinsic gene expression noise by pulsatile TF activity[23]; however, the exact source and the requirements for such noise reduction remain to be studied in more detail. Our results suggest the possibility that variability reduction may be a functional role of pulsing in cellular signaling[5]. In contrast, Hansen and O'Shea found that information transmission via frequency modulated signaling is less reliable than AM signalling for the stress-responsive yeast TF *Msn2*[39]. This discrepancy may result in part from differences in the dynamics of the applied inputs. However, it may also result from differences between natural and synthetic promoters[40] as well as the activity of the *Msn2* and the viral VP16 activation domain, which could both affect the transcriptional response to pulsed inputs. It thus remains to be seen whether pulsing-mediated variability attenuation can be identified in natural systems.

We found that PWM can be employed for proteins with relatively short half-lives (Fig. 1g). At the same time, protein (and mRNA) stability sets a limit on relevant PWM periods (Fig. 1g, Supplementary Note 2). A consequence of this trade-off is that the benefits of PWM are less pronounced at low induction levels that require the use of short input pulses (see for example the variability reduction for a 10% duty cycle in Fig. 3a). In comparison, a synthetic feedback circuit that does not suffer from this complication was shown to enable expression with low variability over very wide ranges of expression levels[14]. One possible way to further extend the functional range of

PWM would be to use an optogenetic TF with an even faster dark-reversion rate[41] (Supplementary Note 2) or an expression system whose activity can be regulated bidirectionally with two separate inputs[42]. Notably, the mechanisms behind the benefits of pulsatile regulation are not specific to VP-EL222 and should be widely applicable to systems based on fast-acting regulators in a variety of organisms. For example, attenuation of TF variability may be important for the precise and graded control of endogenous transcription using light-inducible CRISPR-Cas9 systems[43] in mammalian cells, where transient transfections are often performed.

## Methods

**Plasmid construction.** *E. coli* TOP10 cells (Invitrogen) were used for plasmid cloning and propagation. Sequences and details of all DNA constructs used in this study can be found in Supplementary Note 5. All plasmids used in this study are summarized in Supplementary Table 1. Plasmids were constructed by restriction-ligation cloning using enzymes from New England Biolabs (USA).

All PCRs were performed using Phusion Polymerase (New England Biolabs). The EL222-based transcription factor under control of the *ACT1* promoter was cloned in an integrative vector based on the pRS vector series[44] and a low-copy plasmid (pRG215)[9]. Constructs with light-inducible promoters were cloned in pFA6a-*His3MX6*[45]. All constructs were verified by sanger sequencing (Microsynth AG, Switzerland).

**Yeast strain construction.** All strains are derived from BY4741 and BY4742 (Euroscarf, Germany)[46]. All strains used in this study are summarized in Supplementary Table 2. Transformations were performed with the standard lithium acetate method[47] and selection was performed on appropriate selection plates. The basis of the majority of strains used in this study are DBY41 and DBY42. Both strains express VP-EL222 from the *ACT1* promoter and were generated by transforming PacI digested plasmid pDB58 into BY4741 and BY4742, respectively. Plasmid integration was verified by function. All light-inducible promoter constructs were PCR amplified using primers for the integration into the *HIS3* locus (Primers HIS3_integration_fwd/rv, Supplementary Table 3). Integration of reporter constructs was verified via PCR and function. Promoter sequences of strains shown in Fig. 2a were verified by sequencing. Reference strains (Supplementary Note 4) were constructed by tagging the respective proteins (Supplementary Table 5) in BY4741 with mCitrine. PCR amplicons used for transformation were amplified from a mCitrine containing pFA6a-*His3MX6*-based plasmid[45] using primers presented in Supplementary Table 3. Diploid strains were generated by mating and selection by growth on SD plates lacking both L-Lysine and L-Methionine.

**Media and growth conditions.** All experiments were performed in synthetic medium (SD; LOFLO yeast nitrogen (ForMedium), 5 g/L ammonium sulfate, 2% glucose, pH was adjusted to 6.0). All experiments were performed in 25 ml glass centrifuge tubes (Schott 2160114, Duran) stirred with $3 \times 8$ mm magnetic stir bars (Huberlab) using a setup comprised of a water bath (ED (v.2) THERM60, Julabo) set to 30 °C, a multi position magnetic stirrer (Telesystem 15, Thermo Scientific) set to 900 rpm, a 3D printed, custom-made 15-tube holder, and custom-made LED pads (460 nm peak wavelength) located underneath the culture tubes. A white diffusion filter (LEE Filters) was placed between the LED and the culture tube to allow for even illumination. LED intensity was measured at 4 cm distance from the light source using a NOVA power meter and a PD300 photodiode sensor (Ophir Optronics).

**Flow cytometry.** All experiments except growth and smFISH were performed in the following way.

Cultures were grown overnight starting from single yeast colonies, subcultured in fresh medium and grown for at least 16 h in the dark while maintaining an optical density at 700 nm ($OD_{700}$)[48] lower than 0.4. At the start of the experiment, cells were diluted to an $OD_{700}$ of 0.005 in 4 ml of medium. Before measurement, cell samples were incubated in SD media with 0.1 mg/ml cycloheximide for 3.5 h at 30 °C to ensure full fluorescent protein maturation. The maturation step was omitted for a destabilized version of the Citrine FP. Samples were analyzed using a LSRFortessa™ LSRII cell analyzer (BD Biosciences, Germany). To measure mKate2 fluorescence, a 561 nm excitation laser and a 610/20 nm emission filter and for mCitrine, a 488 nm excitation laser and a 530/11 nm emission filter were used. Data were analyzed using R (3.3.2) with the flowCore package[49]. Cells were gated based on forward and side scatter to remove debris and cell aggregates. For strains containing centromeric plasmids, a budded cell population was selected by gating based on the forward and side scatter width[50]. We found that this population shows a higher percentage of responsive cells, which likely results from a higher degree of plasmid retention (Supplementary Fig. 12c,d). Strong outliers were removed from the data as follows: First, the fluorescence values were log-transformed. Outliers were defined as data points with an absolute deviation from the fluorescence distribution median of greater than threefold of the median absolute deviation.

For the analysis of gene expression heterogeneity, fluorescent levels were normalized by side scatter area to reduce the effect of cell size (see Supplementary Fig. 9)[51]. At least 1000 cells and typically 5000 cells were analyzed.

**Fluorescence activated cell sorting and plating assay.** For strains containing centromeric VP-EL222 plasmids, we found that a subpopulation of yeast cells defined based on forward and side scatter showed a higher percentage of light-responsive cells than the whole population. To analyze the origin of the difference in responsiveness, we compared plasmid retention/*LEU2* marker expression in these population. Cells (strains DBY112 (VP-EL222 on centromeric plasmid) and DBY43 (stable integration of VP-EL222)) were grown as described above in SD medium lacking L-Leucine (SD -LEU) for the flow cytometry experiments. Cells were then sorted based on forward and side scatter (Supplementary Fig. 12c) into 4 ml of fresh SD -LEU media (40,000 cells per gate) using a BD Influx™ cell sorter (BD Biosciences, Germany). Cells were then diluted to a cell concentration of 1000 cells per ml and 100 μl were plated on SD and SD -LEU plates. After 3 days of growth, plate images were acquired and colonies were counted using the "Spot Detector" plugin in Icy[52,53] (see Supplementary Fig. 12d for results).

**Modeling.** The ODE modeling and parameter fitting is described in detail in Supplementary Note 1 (see Supplementary Table 4 for model parameters). The model consists of the following three ordinary differential equations describing VP-EL222 activation (1), VP-EL222 dependent mRNA expression (2), and protein expression (3):

$$\frac{d\text{TF}_{on}}{dt} = I \cdot k_{on} \cdot (\text{TF}_{tot} - \text{TF}_{on}) - k_{off} \cdot \text{TF}_{on}, \quad (1)$$

$$\frac{d\text{mRNA}}{dt} = k_{basal} + k_{max} \cdot \frac{\text{TF}_{on}^n}{K_d^n + \text{TF}_{on}^n} - k_{degR} \cdot \text{mRNA}, \quad (2)$$

$$\frac{d\text{Protein}}{dt} = k_{trans} \cdot \text{mRNA} - k_{degP} \cdot \text{Protein}, \quad (3)$$

Simulations and model fitting were performed using Matlab (R2014a, Mathworks).

Stochastic/stochastic hybrid modeling is described in detail in Supplementary Note 4. Due to high copy numbers of the reporter protein, we used a hybrid modeling approach in which reporter protein translation and degradation was simulated using an ODE and all other reactions were simulated using the stochastic simulation algorithm (SSA)[54]. We used a custom implementation (provided by Jan Mikelson, ETH Zürich) of the SSA simulation combined with an ODE solver (CVODE package for C++). Since the stochastic part does not depend on the continuous dynamics, the simulation performed the common SSA steps for the stochastic part and reinitialized the ODE solver after each reaction to compute the deterministic dynamics in between the stochastic reactions.

**Dual-reporter experiments.** Dual-reporter experiments were performed using the diploid strain DBY110. This strain was constructed by mating DBY43 and DBY104, expressing mKate2 and mCitrine from 5xBS-*CYC180*pr integrated into the *HIS3* locus. The equivalence of both reporter genes is shown in Supplementary Fig. 10. mCitrine fluorescence values were adjusted by multiplication with a constant in order to equate the mean values of mCitrine and mKate2 fluorescence measurements. Using the formalism introduced in ref. [24], total variability was decomposed into extrinsic and intrinsic variability using the following equations:

$$\text{CV}_{int}^2 = \frac{\langle (r - y)^2 \rangle}{2 \langle r \rangle \langle y \rangle}, \quad (4)$$

$$\text{CV}_{ext}^2 = \frac{\langle ry \rangle - \langle r \rangle \langle y \rangle}{\langle r \rangle \langle y \rangle}, \quad (5)$$

Here, $r$ and $y$ are vectors whose elements are cellular fluorescence values for mKate2 and mCitrine, respectively. Angled brackets represent population means.

**Measuring the influence of light on cell growth.** Cells were initially grown as described for flow cytometry experiments. At the start of the experiment, cultures were diluted to an $\text{OD}_{700}$ of 0.01 in a total volume of 6 ml. Cells were grown for 2 h before starting blue-light illumination. Subsequently the $\text{OD}_{700}$ was measured every hour for 6 h and the growth rate was calculated by performing linear regressions on log-transformed OD-data. Growth curves and analysis results are show in Supplementary Fig. 1.

**Measuring the effect of URA3 on cell growth.** In order to measure how Ura3p expression affects cell growth, DBY125 expressing both mCitrine and Ura3p from two separate 5xBS-*CYC180* promoters was initially grown as described for all other flow cytometry experiments (see above). Cells were then diluted to an OD of 0.0001

in SD medium and illuminated for 14 h under the following light conditions. AM: 56, 77, 98, 119, 133, 162 μW cm$^{-2}$ light intensity. PWM: 420 μW cm$^{-2}$ light intensity; 30 min period; 3.33, 6.66, 10, 13.33, 20, 33.33% duty cycle.

Cells were then washed with SD medium lacking L-uracil and resuspended in 4 ml of this medium. Cells were further grown under the same illumination conditions and samples were analyzed every hour by measuring the cell count per 60 μl medium using a CytoFLEX flow cytometer (Beckman Coulter) for 5 h starting 1 h after washing. Growth rates were calculated by performing linear regressions on log-transformed count-data. Fluorescence measurements were performed after the washing step using flow cytometry as described above.

**Single molecule FISH experiments.** For single molecule FISH experiments, DBY89 was grown from a single colony to saturation in SD medium. Cultures were diluted to reach an $\text{OD}_{700}$ of 0.4 at the start of the experiment the next day. For each time point, 4 ml cell culture were transferred to 25 ml glass centrifuge tubes stirred with $3 \times 8$ mm magnetic stir bars. Illumination was performed with a light intensity of 350 μW cm$^{-2}$ for 20 min.

Cell fixation and probe hybridization was performed as described previously[55]. Briefly, after 0, 10, 20, 30, 40, and 60 min (where 0 min marks the start of illumination), cells were fixed for 45 min after adding 400 μl of 37% formaldehyde (Sigma-Aldrich) to the culture medium. Spheroplasting was performed using a final Lyticase (Sigma-Aldrich) concentration of 50 units/ml. The progression of spheroplasting was monitored under the microscope. Cells were stored in 70% ethanol at 4 °C overnight. Hybridization was performed using multiple probes complementary to the PP7 SL and singly labeled with CY3 at a 0.1 μM concentration (synthesized by Integrated DNA Technologies, sequences are listed in Supplementary Table 3)[56]. Cells were stained with DAPI (0.1 μg/ml in PBS, Sigma-Aldrich), attached to Poly-D-Lysine treated coverslips, and slips were mounted on slides using Prolong Gold mounting medium (Invitrogen).

**Microscopy setup.** All images were taken with a Nikon Ti-Eclipse inverted microscope (Nikon Instruments), equipped with a Plan Apo Lambda ×100 Oil objective (Nikon Instruments), Spectra X Light Engine fluorescence excitation light source (Lumencor, USA), pE-100 brightfield light source (CoolLED Ltd., UK), and CMOS camera ORCA-Flash4.0 (Hamamatsu Photonic, Switzerland). The camera was water-cooled with a refrigerated bath circulator (A25 Refrigerated Circulator, Thermo Scientific). The microscope was operated using NIS-Elements software. Z-stacks consisting of 31 images with a step size of 0.1 μm were taken for CY3 (Excitation: 542/33, Emission: 595/50) and DAPI (Excitation: 390/22, Emission: 460/50). Phase contrast images were taken at the reference point.

**Microscopy image analysis.** The image analysis procedure was performed using custom Matlab scripts and consists of three steps: segmenting individual nuclei (based on DAPI images), locating fluorescent spots in the nuclear regions, and quantifying the intensity of these spots.

Nuclei were first enhanced by using the Difference of Gaussians algorithm. Nuclear regions were then segmented by manually optimized thresholding. Detected regions that were too big or small to represent nuclei were removed. For each nuclear region, a Difference of Gaussian algorithm was used to enhance spots in the CY3 images and spots were identified using thresholding. In order to quantify the intensity of the nuclear spots, the sum of a two-dimensional Gaussian function and a 2D-plane was fitted in a square area around the identified spot with an edge length of 19 pixels. If no spot was detected, the same procedure was performed at the center of the nuclear region. Spot intensity was then defined as the integral of the Gaussian function. For each nucleus/cell, the spot with the highest intensity was defined as the transcription site.

**Data availability.** Data, plasmids, strains are available from the corresponding author upon request.

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

## Acknowledgements

We thank for Peter Buchmann and Marc Rullan for the construction of and help with the LED-setup, Verena Jäggin for assistance with FACS, and Jan Mikelson for providing software for stochastic simulations. We thank members of the Khammash lab, Serge Pelet, and Martin Ackermann for helpful discussion and Stephanie Aoki for critical reading of the manuscript. We would further like to thank Fabian Rudolf (ETH Zurich), Kevin Gardner (City University of New York), Robert H Singer (Albert Einstein College of Medicine), and Daniel Zenklusen (Université de Montréal) for providing plasmids. D. B. is part of the Life Science Zurich graduate school. This project has received funding from the European Research Council (ERC) under the European Union's Horizon 2020 research and innovation programme grant agreement no. 743269 (CyberGenetics project).

## Author contributions

D.B. conceived the study, performed experiments, and mathematical modeling, analyzed data, and wrote the paper. M.K. supervised the study, analyzed data, secured funding, and wrote the paper.

**Additional information**

**Competing interests:** The authors declare no competing interests.

