## [Peer Review File · Nature Communications]

Reviewers' comments:

Reviewer #1 (Remarks to the Author):

In this article, the authors utilize pulse-width modulation (PWM) as opposed to amplitude modulation (AM) of the light-activated transcription factor EL222 (Motta-Mena, et al 2014) to control gene expression. They show that this "pulsatile" mode of gene regulation, which is reminiscent of the pulsatile TF activity seen for many natural yeast transcription factors (Cai, et al 2008; Dalal, et al 2014) leads to (1) reduced cell-to-cell variability in protein expression (2) independent tuning of protein expression mean and variability and (3) graded multi-gene regulation at fixed expression ratios. These results are exciting, both for understanding natural pulsatile regulation of transcription factors but perhaps more importantly for suggesting powerful ways of controlling protein expression in synthetic genetic circuits.

The authors first characterize the EL222 expression system in *Saccharomyces cerevisiae* using a core Cyc1 promoter with 5 binding sites for EL222 driving expression of mKate. A kinetic model of the system is fit to three characterization experiments (gene expression time course, dose-response for AM and PWM). Analysis of the model shows that, unsurprisingly, fast deactivation of the VP-EL222 system is essential for PWM control. Using the model they infer that the half-life of activated VP-EL222 is <2 minutes. Given this deactivation rate they can predict that PWM will result in protein expression that is relatively constant for a wide range of protein half-lives (Figure 1g). This is predicted, but not shown experimentally except for the stable mKate protein. They conclude that this system can be used with PWM to control protein expression for a wide-variety of proteins with different half-lives.

They authors show that PWM can be used to keep a fixed protein expression ratio when protein expression is driven from promoters with different numbers of binding sites (but see below).

A particularly striking and exciting aspect of this work is showing that pulsatile regulation decreases expression variability. The authors show that both intrinsic (stochastic promoter events) and extrinsic (global differences between cells) noise are reduced using pulsatile stimulation. This is done using the now standard two-color experiments in single-cells. Presumably this noise reduction is due to the PWM scheme operating in low variability regimes of the TF concentration relative to production rate (Figure 3d). Experiments using expression of VP-EL222 from plasmids and showing the correlation between VP-EL222 (Citrine) and mKate expression make a convincing argument that PWM is able to overcome variability in TF concentration (that AM regulation cannot).

Comments and Questions

- Is there some reason that you only demonstrated coordinated expression between the 5X and 2X Cyc promoters? This result would be more convincing if multiple promoters were shown to have coordinated expression under pulsatile stimulation.
- I am somewhat puzzled by the inclusion of the URA3 regulation data. The authors show that PWM modulation of URA3 protein expression leads to better growth rates in uracil limited media than AM modulation, presumably because PWM leads to tighter regulation of URA3 protein expression. However, this is not directly measured (the PCYC1-Citrine promoter is a proxy) and therefore there is not enough evidence to make this claim. This result should either be further explored (URA3 expression can be directly measure over the different PWM modulation schemes) or doesn't need to be included in this study in my opinion.
- When cells carried centromeric plasmids, budded cells were selected by flow cytometry because this population had a higher percentage of responsive cells. The authors posit (methods: Flow Cytometry) that this is due to a higher degree of plasmid retention. This is a likely explanation, but should be verified by comparing resistance/marker presence between the gated and ungated populations. This seems like a straightforward enough experiment, and would justify using the more-responsive population for analysis.
- The authors claim that the system and blue-light exposure do not affect growth rate. While growth rates are shown in Supplemental Figure 1, the curves themselves are not shown and how the growth rate is calculated is not clear. There are several packages and methods for doing this

(e.g. growth) so what the authors are actually using/doing here is important. I'd actually like to see the growth curves in the supplemental because many details can be missed/hidden depending on how the growth rate is calculated.

Reviewer #2 (Remarks to the Author):

This manuscript aims to test gene expression control with pulsatile light. The bacterial light-oxygen-voltage (LOV) EL222 protein fused to an activator domain is used for this purpose. Light induces rapid dimerization and DNA binding of EL222 monomers. Lack of light causes quick reversal to the monomeric state and DNA unbinding. The fast on/off dynamics of this protein enables light-controllable Pulse Width Modulation (PWM) within fixed On/Off periods. Compared with Amplitude Modulation (AM) reveals that PWM can achieve a similarly large dynamic range, but its dose-response is linear as a function of the duty cycle. Moreover, gene expression variability decreases with the PWM period at various duty cycles, enabling decoupled control of gene expression mean and noise via a single gene circuit in genetically identical cell populations. Computational modeling reveals that lower PWM noise requires saturating transcription factor concentrations, such that gene expression fluctuations are not transmitted through the regulatory cascade.

These results are novel and exciting. The work is carefully executed and clearly presented. The computational model provides interesting insights. Overall, this is an outstanding interdisciplinary manuscript that deserves to be published, once the following comments have been addressed.

Major comments:

(1) While the pulsatile aspect of gene expression control is similar to the frequency-modulated transcriptional regulation investigated by the Elowitz lab, the latter involves shuttling in/out of the nucleus. How far does the analogy go considering this molecular-mechanistic difference? Under what assumptions are these two mechanisms (subcellular shuttling versus dimerization) equivalent? When would they be different? This would be worth discussing in a couple of sentences.

(2) Considering the known toxicity of the VP16 domain, besides showing the growth rates, it would be important to show the growth curves in Figure S1. How exactly were the growth rates calculated? Another potential source of toxicity is illumination itself. In fact, pulsatile illumination has been widely used in the field, exactly to avoid toxicity. How does the light intensity here compare to the levels reported to be toxic elsewhere in the literature? Low illumination intensity is the likely reason why no phototoxicity was seen here. For comparison, phototoxicity in fibroblasts is seen already at light energies over 110 J/cm².

(3) PWM is always applied switching between the maximum (saturating) intensity and no light. How would the conclusions change (especially the variability) if PWM is applied between intermediate & zero, or intermediate and full light intensity?

(4) The references could benefit from some clean-up. Specifically, in lines 125-132, reference 17 cited for TATA box modification, although no TATA box mutations are described in that reference. An appropriate reference for TATA-box based noise control would be PMID:17189188. On the other hand, reference 17 (PMID:19279212) would be important to cite for feedback-based variability regulation.

(5) Another reason why reference 17 (PMID:19279212) would be important to discuss is because it demonstrates linear tuning of gene expression with the level of inducer in yeast. Exactly to the point in the Introduction, "graded gene expression regulation allows us to obtain a quantitative

understanding of the expression-phenotype mapping". This earlier gene circuit should be mentioned when linearity versus the duty cycle is described in Figure 2C. Overall, the PWM results should be evaluated and compared with this earlier TetR-based gene circuit that has been called the "linearizer". For example, the linearizer was able to tune gene expression up to saturation with low noise at all induction levels. In contrast, noise increases with PWM control as expression decreases. The linearizer maintained linearity over 3 decades of expression levels – how does this compare to PWM? It would be interesting to see the noise-mean characteristics in Fig. 3A for lower mean levels, possibly log-scaling the horizontal axis. Also, the dose-response is not always linear versus the duty cycle (Figs. S2C, S5B). What are the conditions for PWM dose-response linearity?

(6) While the model successfully reproduced the CV versus the mean in Fig. 3E, this model is entirely based on extrinsic noise. On the other hand, Figures 3B and 3C indicate that extrinsic noise does not dominate here – actually, the intrinsic and extrinsic variabilities are quite comparable. So a model producing both types of variabilities would be most appropriate. This may be worth acknowledging briefly. Also, the classical way of calculating intrinsic and extrinsic noise has some limitations, see PMID:22225397.

(7) In Figure 3A, the CV decreases with the PWM period. This is somewhat counterintuitive if noise reduction originates from stable reporter proteins low-pass filtering rapid promoter-level fluctuations. A possible simple intuitive explanation why faster fluctuations result in higher downstream noise could be based on Figure 3D, that TF activity does not take on the extreme values so much and therefore it starts spending more time in the middle and resembling the AM with the same average. Perhaps this should be stated.

(8) Fig 1: Rate limitation is not really based on period of PWM, but rather on kinetics of system, or more specifically how well the system tracks your light input. From figure 1f, it's apparent that the maximum tracking rate is going to be around 50% of 7.5min or 3.25min since it takes around 3.25 min to reach peak expression after light induction. This is probably why the 30min period worked well for a minimum duty cycle of 10% (around 3 minutes), but really it's not the duty cycle per se that's the limitation, it's the actual amount of time of light induction (min approx 3min). As an example, a period of 15 minutes would probably be fine for a 50% duty cycle (as shown in figure 1f). Would noise increase for even smaller duty cycles than the ones shown?

(9) Regarding the metabolic burden in line 167, it may be worth also mentioning that metabolic burden can result in quick evolutionary degradation of activator-based synthetic gene circuits, see PMID:26324468.

(10) The "tracking score" is hard to interpret and does not really describe the increase in baseline well. In fact, fig 1f doesn't really show that the baseline increases over time. Better would be to fit a line to the minimum points in each cycle and report the slope. Perhaps use covariance between light (input) and fluorescence (output) as an alternative?

Minor comments:

(11) The title is a bit too long. It would be great if it could be shortened somehow.

(12) Figure 1D: Probably not essential, but the control case with the transactivator and without reporter could make this figure complete.

(13) Figure 1E: The denominator for RNA production should probably include the $k_d + TF_{on}$, rather than the product?

(14) Figure 1G How does mKate expression look before 360 min?

(15) Figure 2A: How would a pCYC180 construct with a single EL222 binding site behave? Perhaps

discuss.

(16) 'However, the ability to tune variability may allow for the analysis of its phenotypic consequences.' What meant by "tuning variability"? Typically the requirement is to alter variability while keeping the mean fixed. See, for example, PMID:17189188.

(17) Figure 3H: May be worth explicitly stating that URA3 is needed for survival, therefore cells not expressing it in a uracil-free media will die. How many cells are in Figure 3h?

(18) Discussion: 'We further uncover a novel mechanism for noise reduction and tuning in gene expression systems based on pulsatile inputs.' Change mechanism to "method" or another synonym.

(19) Materials & Methods: Were any of the yeast strain constructions verified via sequencing in addition to function?

(20) Flow cytometry: 'At least 1000 cells per sample were analyzed.' That is a lower bound - how many cells typically?

(21) Supplement Figure 5, fix the legend labeled 'b' & 'c' to 'a' & 'b'

(22) Supplement Figure 12, please place a legend on the graph.

(23) Fig. S3A: CY3 and mKate2 might have some spectral overlap. For example, at typical 555nm excitation for Cy3, mKate2 is about 50% excited. Depending on selectivity of emission filter, there can be significant cross-talk between CY3 and mKate2. Was this possibility checked and addressed?

Reviewer #3 (Remarks to the Author):

In this paper, the authors demonstrated the advantages and applications of dynamic pulsatile signals for gene expression. Specifically, using an optogenetic transcriptional control system, they showed that dynamic pulsatile signals through pulse-width modulation (PWM) can reduce cell-to-cell variability. By encoding dynamic signals into a single input, they also demonstrated that the mean and variability of gene expression can be precisely and independently tuned. They further created a promoter library to illustrate graded, multi-gene control of gene expression at fixed expression ratios using pulsatile signaling. These properties of the transcriptional control strategy hold both theoretical and practical values for the fields of synthetic biology and systems biology. The manuscript is also clearly written and easy to follow. However, the current version of the manuscript is incomplete, and several main issues need to be addressed.

Major points:

1. The key message the paper tried to convey is that PWM is superior than amplitude modulation for gene expression control and such a strategy can be applied for synthetic biology purposes. Toward that end, I feel that the paper is not a complete story due to the lack of a compelling application example that can showcase the utility of such a controlling concept.

2. Related to the above comment, in page 4 (line 162) the authors tried to illustrate the application of PWM for metabolic gene control. However, the data provided here is insufficient as a proof of concept: In an analysis where variability is the major factor, two repeats is far from sufficient to draw a sound conclusion. In addition, in Fig. 3h it can also be seen that there is considerable overlap between the two cases, and the separation of the two lines is only possible after taking the average. Thus, while this is an interesting topic, the current version of the

manuscript lacks a convincing example to make the case.

3. In the study, only a single protein with a long half-life was used to demonstrate that the in-period fluctuation is small. However, to prove the point, it would be persuasive to examine proteins with reduced stabilities. Importantly, it will be interesting to see if proteins with short half lives, which are intrinsically noisy, continue to have reduced fluctuations. If that is the case, it would have broad applications in different settings.

4. One of the major claims of this article is that gene expression variability can be tuned ("precisely", as stated in the abstract). This point isn't sufficiently shown by the results though. What is shown by the data (Fig. 3a, e) is that different periods of PWM result in different variability in a nonlinear way, which doesn't give much flexibility in achieving a specific variability by tuning the period in a quantitative way.

5. As for the more fundamental exploration of the mechanism, the reduction of variability by PWM is analyzed compared to AM, but the mechanism about how variability can be tuned by PWM, namely, how changing the period can lead to changes in variability, is lacking. There is thus a mismatch between the major claim, and the main efforts in the analysis provided. I would suggest the authors to examine how CV correlates with PWM period in a quantitative way with simulations. It will also be valuable to provide a mechanism to explain why a longer period results in a lower variability. Additionally, a more "precise" modeling approach (stochastic model) can be a more theoretically sound framework to study the variability for the experimental systems.

Minor points:

1. Fig. 1e: it seems that the rate of mRNA production is incorrect.

We thank all reviewers for their in-depth review and helpful comments. Based on these comments we have made a series of new experiments and analyses and clarified several points throughout the text. In particular, we have:

- Performed experiments using a destabilized protein to show that PWM can also be employed with such proteins.
- Improved upon the *URA3* expression experiment by directly tagging *URA3* with a fluorescent protein and using a different VP-EL222 responsive promoter that allows us to map the effects of expression variability on cell growth over wider ranges of expression levels
- Performed stochastic modeling to identify a relatively simple model that can quantitatively recapitulate both intrinsic and extrinsic contributions to expression variability for AM and PWM
- Extended our analysis on co-regulation / “dose-response-linearization” to additional promoters
- Extended the discussion section of the manuscript

Below we provide point-to-point responses to the comments of the reviewers. Our responses as well as all changes made to the main text and Supplement are highlighted in blue.

Reviewers' comments:

Reviewer #1 (Remarks to the Author):

In this article, the authors utilize pulse-width modulation (PWM) as opposed to amplitude modulation (AM) of the light-activated transcription factor EL222 (Motta-Mena, et al 2014) to control gene expression. They show that this “pulsatile” mode of gene regulation, which is reminiscent of the pulsatile TF activity seen for many natural yeast transcription factors (Cai, et al 2008; Dalal, et al 2014) leads to (1) reduced cell-to-cell variability in protein expression (2) independent tuning of protein expression mean and variability and (3) graded multi-gene regulation at fixed expression ratios. These results are exciting, both for understanding natural pulsatile regulation of transcription factors but perhaps more importantly for suggesting powerful ways of controlling protein expression in synthetic genetic circuits.

The authors first characterize the EL222 expression system in *Saccharomyces cerevisiae* using a core *Cyc1* promoter with 5 binding sites for EL222 driving expression of mKate. A kinetic model of the system is fit to three characterization experiments (gene expression time course, dose-response for AM and PWM). Analysis of the model shows that, unsurprisingly, fast deactivation of the VP-EL222 system is essential for PWM control. Using the model they infer that the half-life of activated VP-EL222 is <2 minutes. Given this deactivation rate they can predict that PWM will result in protein expression that is relatively constant for a wide range of protein half-lives (Figure 1g). This is predicted, but not shown experimentally except for the stable mKate protein. They conclude that this system can be used with PWM to control protein expression for a wide-variety of proteins with different half-lives.

They authors show that PWM can be used to keep a fixed protein expression ratio when protein expression is driven from promoters with different numbers of binding sites (but see below).

A particularly striking and exciting aspect of this work is showing that pulsatile regulation decreases expression variability. The authors show that both intrinsic (stochastic promoter events) and extrinsic (global differences between cells) noise are reduced using pulsatile stimulation. This is done using the now standard two-color experiments in single-cells. Presumably this noise reduction is due to the PWM scheme operating in low variability regimes of the TF concentration

relative to production rate (Figure 3d). Experiments using expression of VP-EL222 from plasmids and showing the correlation between VP-EL222 (Citrine) and mKate expression make a convincing argument that PWM is able to overcome variability in TF concentration (that AM regulation cannot).

Comments and Questions

- Is there some reason that you only demonstrated coordinated expression between the 5X and 2X Cyc promoters? This result would be more convincing if multiple promoters were shown to have coordinated expression under pulsatile stimulation.

This was mainly a result of the relatively low throughput of the experimental setup used in this study. We have now also measured AM and PWM dose-response curves for additional promoters. The results are presented in **Fig. 2e** as well as **Supplementary Fig. 7b-e**. We show now that both tuning of TF-binding site number as well as the use of different promoter backbones can result in differences in dose response curves that can be attenuated by using PWM.

- I am somewhat puzzled by the inclusion of the URA3 regulation data. The authors show that PWM modulation of URA3 protein expression leads to better growth rates in uracil limited media than AM modulation, presumably because PWM leads to tighter regulation of URA3 protein expression. However, this is not directly measured (the PCYC1-Citrine promoter is a proxy) and therefore there is not enough evidence to make this claim. This result should either be further explored (URA3 expression can be directly measure over the different PWM modulation schemes) or doesn't need to be included in this study in my opinion.

We have now improved upon the experiments in two main ways. We have directly labeled URA3 with mCitrine and thus we no longer rely on a proxy reporter. Furthermore, we have changed the VP-EL222 responsive promoter used to drive the expression of URA3. This enabled us to measure a more complete dose-response curve which now more clearly shows that the dose-response curve between URA3 expression and growth differs between AM and PWM (**Fig. 5**). It further uncovers a beneficial effect of noise at low expression levels and still shows that PWM enables maximal growth at lower expression levels. We further use a simple model to illustrate how these observations can result from sharp dose-response curves / all-or-none behaviour on the single cell level.

- When cells carried centromeric plasmids, budded cells were selected by flow cytometry because this population had a higher percentage of responsive cells. The authors posit (methods: Flow Cytometry) that this is due to a higher degree of plasmid retention. This is a likely explanation, but should be verified by comparing resistance/marker presence between the gated and ungated populations. This seems like a straightforward enough experiment, and would justify using the more-responsive population for analysis.

We have explored the topic by sorting cells of the gated population and comparing colony counts on plates containing or lacking L-Leucine (*LEU2* is the auxotrophy marker used for plasmid selection). The results suggest a higher degree of plasmid retention in the population used for analysis (**Supplementary Fig. 11c**).

- The authors claim that the system and blue-light exposure do not affect growth rate. While growth rates are shown in Supplemental Figure 1, the curves themselves are not shown and how the growth rate is calculated is not clear. There are several packages and methods for doing this (e.g. grofit) so what the authors are actually using/doing here is important. I'd actually like to see the growth curves in the supplemental because many details can be missed/hidden depending on how the growth rate is calculated.

Given that both phototoxicity as well as VP-EL222 activation could potentially affect cell growth, we intended to measure the growth behaviour under the same illumination conditions as used in the reported experiments. Given our experimental setup, it is however not possible to measure growth curves over extended periods of time. We have thus measured growth rates under the same conditions in which we perform our experiments, meaning exponential growth over a period of 6h. We calculate the growth rate by performing a linear regression on log-transformed OD data (measured by sampling once an hour). We have now added the growth curves to Supplementary Fig. 1a. To state the case more accurately, we now write “Neither blue illumination nor VP-EL222 activation affected cell growth or constitutive gene expression under the experimental condition and timescales”. Further evidence that light exposure does not significantly affect growth rate can be seen in Supplementary Fig. 13b), which shows results of an experiment where cells were grown for 14 h before starting growth measurements. In these experiments, we find that illumination does not affect cell growth, again the growth rates were calculated as described above.

Reviewer #2 (Remarks to the Author):

This manuscript aims to test gene expression control with pulsatile light. The bacterial light-oxygen-voltage (LOV) EL222 protein fused to an activator domain is used for this purpose. Light induces rapid dimerization and DNA binding of EL222 monomers. Lack of light causes quick reversal to the monomeric state and DNA unbinding. The fast on/off dynamics of this protein enables light-controllable Pulse Width Modulation (PWM) within fixed On/Off periods. Compared with Amplitude Modulation (AM) reveals that PWM can achieve a similarly large dynamic range, but its dose-response is linear as a function of the duty cycle. Moreover, gene expression variability decreases with the PWM period at various duty cycles, enabling decoupled control of gene expression mean and noise via a single gene circuit in genetically identical cell populations. Computational modeling reveals that lower PWM noise requires saturating transcription factor concentrations, such that gene expression fluctuations are not transmitted through the regulatory cascade.

These results are novel and exciting. The work is carefully executed and clearly presented. The computational model provides interesting insights. Overall, this is an outstanding interdisciplinary manuscript that deserves to be published, once the following comments have been addressed.

Major comments:

(1) While the pulsatile aspect of gene expression control is similar to the frequency-modulated transcriptional regulation investigated by the Elowitz lab, the latter involves shuttling in/out of the nucleus. How far does the analogy go considering this molecular-mechanistic difference? Under what assumptions are these two mechanisms (subcellular shuttling versus dimerization) equivalent? When would they be different? This would be worth discussing in a couple of sentences.

Based on the fact that our simple ODE modeling approach—in which solely VP-EL222 activation and not dimerization is modeled explicitly—can recapitulate our experimental data, we would expect that our results would also be applicable to the shuttling mechanism or other ways of activation. The activation step could for example be reinterpreted as a light-regulated change in nuclear localization rate.

However, we would also caution that there are many potential intricacies that could lead to different and potentially unexpected outcomes. In fact, even modeling the dimerization process itself without further biochemical information opens up different modeling possibilities (for example: How do dimers degrade?). Given the large number of unknowns / possibilities for even

such simple processes, we feel that we would not be able to compare the two mechanism satisfactorily (other than the rather trivial comment in the paragraph above), and would thus rather not address this topic in more detail in the paper.

(2) Considering the known toxicity of the VP16 domain, besides showing the growth rates, it would be important to show the growth curves in Figure S1. How exactly were the growth rates calculated? Another potential source of toxicity is illumination itself. In fact, pulsatile illumination has been widely used in the field, exactly to avoid toxicity. How does the light intensity here compare to the levels reported to be toxic elsewhere in the literature? Low illumination intensity is the likely reason why no phototoxicity was seen here. For comparison, phototoxicity in fibroblasts is seen already at light energies over 110 J/cm².

Given that both phototoxicity as well as VP-EL222 activation could potentially affect cell growth, we intended to measure the growth behaviour under the same illumination conditions as used in the reported experiments. However, for the experimental setup, it is not possible to measure “full” growth curves over extended periods of time. We have thus measured growth rates under the same conditions in which we perform our experiments, meaning exponential growth over a period of 6h. We calculate the growth rate by performing a linear regression on log-transformed OD data (measured by sampling once an hour, as described in the Methods section). We have now added the growth curves to Supplementary Fig. 1 (need to add). To not make any overstatements, we now write “Neither blue illumination nor VP-EL222 activation affected cell growth or constitutive gene expression under the experimental condition and timescales”. Further evidence that light exposure does not significantly affect growth rate can be seen in **Supplementary Fig. 13b** which shows results of an experiment where we grow cells for 14 h before starting growth measurements. In these experiments, we find that illumination does not affect cell growth, again the growth rates were calculated as described above. When compared to other studies that investigate the effect of blue light on cellular stress and growth, the light intensities used in our study are relatively low. For example, in (Bodvard et al. 2011) only minor induction of the general stress response is observed for continuous blue light intensities of 3700 $\mu\text{W} / \text{cm}^2$ (at 470 nm), while we use a maximal intensity of 420 $\mu\text{W} / \text{cm}^2$.

(3) PWM is always applied switching between the maximum (saturating) intensity and no light. How would the conclusions change (especially the variability) if PWM is applied between intermediate & zero, or intermediate and full light intensity?

We have now explored the effect of switching between intermediate and zero light inputs experimentally. We found that the resulting variability lies between AM and PWM performed with maximal inputs (Supplementary Fig 12e). This result is also recapitulated by our stochastic model (see below for more information on the modeling).

(4) The references could benefit from some clean-up. Specifically, in lines 125-132, reference 17 cited for TATA box modification, although no TATA box mutations are described in that reference. An appropriate reference for TATA-box based noise control would be PMID: 17189188. On the other hand, reference 17 (PMID: 19279212) would be important to cite for feedback-based variability regulation.

We have now corrected the errors regarding the references in lines 125-132 and cite “PMID: 17189188” for noise tuning by TATA box mutations.

(5) Another reason why reference 17 (PMID: 19279212) would be important to discuss is because it demonstrates linear tuning of gene expression with the level of inducer in yeast. Exactly to the point in the Introduction, “graded gene expression regulation allows us to obtain a quantitative understanding of the expression-phenotype mapping”. This earlier gene circuit should be mentioned when linearity versus the duty cycle is described in Figure 2C. Overall, the PWM results should be evaluated and compared with this earlier TetR-based gene circuit that has been called

the “linearizer”. For example, the linearizer was able to tune gene expression up to saturation with low noise at all induction levels. In contrast, noise increases with PWM control as expression decreases. The linearizer maintained linearity over 3 decades of expression levels – how does this compare to PWM? It would be interesting to see the noise-mean characteristics in Fig. 3A for lower mean levels, possibly log-scaling the horizontal axis. Also, the dose-response is not always linear versus the duty cycle (Figs. S2C, S5B). What are the conditions for PWM dose-response linearity?

Indeed, the “linearizer” is highly relevant and we now also mention the circuit when we write about the observed linear dose-response behavior. Regarding the deviation from linearity shown in Figure S2c and S5b: We typically find that non-linear responses to the duty-cycle occur for low PWM periods / short input pulses. We found that the simple gene expression model recapitulates these observations (qualitatively), suggesting that this behaviour at least partially results from TF activity deviating from a purely pulsatile activity regime at low PWM periods / for short pulses. Another possible explanation for this behaviour may be promoter-dependent differences in the expression response to short TF pulses, which have been observed in natural promoters (Hansen & O’Shea 2013). We do now state these two possibilities in the main text:

“We additionally found that the use of shorter PWM periods resulted in intermediate levels of coordinated promoter regulation, allowing for input-mediated tuning of expression ratios (Fig. 2d). This behavior is qualitatively recapitulated by the simple transcription model (Supplementary Fig. 7e) suggesting that it results from TF activity deviating from a purely pulsatile activity regime at low PWM periods (Fig. 1f, Supplementary Note 1.3). Another possible explanation for this behaviour may be promoter-dependent differences in the expression response to short TF pulses, which have been observed in natural promoters (Hansen & O’Shea 2013).”

Importantly, we find that the promoters we investigated show a (close-to) linear dose response when a 30 min PWM period is used (see also Supplementary Fig. 7).

We now acknowledge the benefit of a genetically encoded feedback regulator, explicitly the TetR-based circuit, for regulating genes at very low variability over wide ranges of expression levels in the discussion section. We did however not perform a quantitative comparison between our results and those presented for the “linearizer” or other circuits as such an analysis can be strongly affected by the experimental and measurement conditions, especially when dealing with fluorescent measurements.

In Fig. 4e, we now provide model predictions for the potential noise behaviour at low expression levels.

(6) While the model successfully reproduced the CV versus the mean in Fig. 3E, this model is entirely based on extrinsic noise. On the other hand, Figures 3B and 3C indicate that extrinsic noise does not dominate here – actually, the intrinsic and extrinsic variabilities are quite comparable. So a model producing both types of variabilities would be most appropriate. This may be worth acknowledging briefly. Also, the classical way of calculating intrinsic and extrinsic noise has some limitations, see PMID:22225397.

We have now also performed simulations using a stochastic model that incorporates both intrinsic and extrinsic / VP-EL222 variability. VP-EL222 variability is modeled by explicitly accounting for its expression dynamics using a simple stochastic gene expression model, meaning that we model the extrinsic noise for the reporter gene as intrinsic noise upstream in the cascade. We found that doing so can nicely recapitulate extrinsic noise in a quantitative manner. We further show that the intrinsic variability can be satisfactorily recapitulated using a standard two-state promoter model. However, we also acknowledge that other models could reproduce the intrinsic behaviour equally well but we cannot make more statements on this using our current data. We perform this analysis by simulating dual color experiments and performing the same analysis / noise decomposition on the model and our experimental data.

The results are summarized in a new paragraph in the main text, Figure (Fig. 4), and Supplementary Note 2.

(7) In Figure 3A, the CV decreases with the PWM period. This is somewhat counterintuitive if noise reduction originates from stable reporter proteins low-pass filtering rapid promoter-level fluctuations. A possible simple intuitive explanation why faster fluctuations result in higher downstream noise could be based on Figure 3D, that TF activity does not take on the extreme values so much and therefore it starts spending more time in the middle and resembling the AM with the same average. Perhaps this should be stated.

We agree with this intuitive explanation and now state this explicitly in the text. We have also added a novel panel (f) to Figure 3 to illustrate this point by showing the temporal changes in TF activity, transcription rate, and transcription variability for different PWM periods and a fixed duty-cycle based on simulations.

(8) Fig 1: Rate limitation is not really based on period of PWM, but rather on kinetics of system, or more specifically how well the system tracks your light input. From figure 1f, it's apparent that the maximum tracking rate is going to be around 50% of 7.5min or 3.25min since it takes around 3.25 min to reach peak expression after light induction. This is probably why the 30min period worked well for a minimum duty cycle of 10% (around 3 minutes), but really it's not the duty cycle per se that's the limitation, it's the actual amount of time of light induction (min approx 3min). As an example, a period of 15 minutes would probably be fine for a 50% duty cycle (as shown in figure 1f). Would noise increase for even smaller duty cycles than the ones shown?

We agree that the kinetics of the system and the actual time of light induction / the pulse-width are the most important characteristics for TF dynamics. The reason why we typically refer to the effects of the PWM periods is that throughout the paper, we compare how different induction regimes affect expression properties for similar gene expression levels and we always keep the period fixed for a given dose response curve. In this context comparing the effects of a 3 min to a 1.5 min pulse requires the comparison of PWM with different periods that achieve a similar mean expression. As seen now in Fig. 4e, we expect noise to further increase with smaller duty cycles.

(9) Regarding the metabolic burden in line 167, it may be worth also mentioning that metabolic burden can result in quick evolutionary degradation of activator-based synthetic gene circuits, see PMID:26324468.

We thank the reviewer for pointing us to this study.

(10) The "tracking score" is hard to interpret and does not really describe the increase in baseline well. In fact, fig 1f doesn't really show that the baseline increases over time. Better would be to fit a line to the minimum points in each cycle and report the slope. Perhaps use covariance between light (input) and fluorescence (output) as an alternative?

As most of our measurements / results are obtained after extended periods of induction (typically 6h), we were interested in quantifying TF dynamics after the mentioned transient increase in baseline. All metrics are calculated after a simulated settling period (see Supplementary Note 1.3). That is why no increase in baseline is observed in the simulated TF dynamics shown in Fig. 1f. We have now added this information to the caption of Figure 1f as it was previously missing. The rationale behind the tracking score is based on the fact that the model does not include any delays. Thus, if the kinetics of TF activity are very fast, it will almost perfectly follow the pulsed light input, which is the desired property, resulting in a "tracking score" of 1. In the extreme case where baseline activity is dominant, the tracking score equals the duty-cycle, giving it a lower bound. We also added this information to the Figure caption to make the score more readily interpretable: "This metric is 1 if the TF activity perfectly tracks the input, which is the desired property, and equals the duty-cycle if TF activity does not change over the PWM period."

Minor comments:

(11) The title is a bit too long. It would be great if it could be shortened somehow.

We have unfortunately not found a satisfactory, shorter title. We have however changed the title and think that it is now more easily accessible, despite its length:
"Pulsatile inputs achieve tunable attenuation of gene expression variability and graded multi-gene regulation"

(12) Figure 1D: Probably not essential, but the control case with the transactivator and without reporter could make this figure complete.

We have now added this additional control to the Figure.

(13) Figure 1E: The denominator for RNA production should probably include the $k_d + TF_{on}$, rather than the product?

We have corrected this error.

(14) Figure 1G How does mKate expression look before 360 min?

For these experiments, we did not perform time-course measurements. We always perform a control that is kept in the dark with all of our experiments, which did not show any significant deviation from the non-induced case shown in Figure 1c.

(15) Figure 2A: How would a pCYC180 construct with a single EL222 binding site behave? Perhaps discuss.

We have added results showing the behavior of the pCYC180 construct with a single binding site to Fig. 2a and Supplementary Fig. S7b,c.

(16) 'However, the ability to tune variability may allow for the analysis of its phenotypic consequences.' What meant by "tuning variability"? Typically the requirement is to alter variability while keeping the mean fixed. See, for example, PMID: 17189188.

Indeed, we meant tuning variability for a fixed mean expression level and have now clarified this in the text.

(17) Figure 3H: May be worth explicitly stating that URA3 is needed for survival, therefore cells not expressing it in a uracil-free media will die. How many cells are in Figure 3h?

We now state this explicitly ("...of the metabolic enzyme Ura3, which is required for survival in uracil-free media..."). At the start of the experiment / after washing, cellular populations are typically around 300,000 cells per 4 ml of culture quantified by counting cells using a flow cytometer.

(18) Discussion: 'We further uncover a novel mechanism for noise reduction and tuning in gene expression systems based on pulsatile inputs.' Change mechanism to "method" or another synonym.

We have changed the wording. The text now reads „ We further propose a novel method for variability reduction and tuning in gene expression systems based on pulsatile inputs"

(19) Materials & Methods: Were any of the yeast strain constructions verified via sequencing in addition to function?

We have now verified the promoter sequences of the reporter constructs shown in Fig. 2a via sequencing.

(20) Flow cytometry: 'At least 1000 cells per sample were analyzed.' That is a lower bound - how many cells typically?

We typically analyzed 5000 cells and have added this information to the methods section. The reason for these relatively low cell counts are that we perform experiments at low cell densities to ensure that conditions are stable during the experiment. We found that 5000 cells give robust results for both mean expression levels and expression variability as quantified by the coefficient of variation.

(21) Supplement Figure 5, fix the legend labeled 'b' & 'c' to 'a' & 'b'

We corrected the legend.

(22) Supplement Figure 12, please place a legend on the graph.

We have made several changes to Supplementary Fig. 12 (now Fig. S13) and also placed a legend on the corresponding panel.

(23) Fig. S3A: CY3 and mKate2 might have some spectral overlap. For example, at typical 555nm excitation for Cy3, mKate2 is about 50% excited. Depending on selectivity of emission filter, there can be significant cross-talk between CY3 and mKate2. Was this possibility checked and addressed?

We agree that the choice of fluorescent dyes for this experiment was not ideal. We have however compared cellular fluorescence levels (in the absence of smFISH spots) in the CY3 channel for cells before and 40 min after the light pulse and found no significant difference between those values.

Reviewer #3 (Remarks to the Author):

In this paper, the authors demonstrated the advantages and applications of dynamic pulsatile signals for gene expression. Specifically, using an optogenetic transcriptional control system, they showed that dynamic pulsatile signals through pulse-width modulation (PWM) can reduce cell-to-cell variability. By encoding dynamic signals into a single input, they also demonstrated that the mean and variability of gene expression can be precisely and independently tuned. They further created a promoter library to illustrate graded, multi-gene control of gene expression at fixed expression ratios using pulsatile signaling. These properties of the transcriptional control strategy hold both theoretical and practical values for the fields of synthetic biology and systems biology. The manuscript is also clearly written and easy to follow. However, the current version of the manuscript is incomplete, and several main issues need to be addressed.

Major points:

1. The key message the paper tried to convey is that PWM is superior than amplitude modulation for gene expression control and such a strategy can be applied for synthetic biology purposes. Toward that end, I feel that the paper is not a complete story due to the lack of a compelling application example that can showcase the utility of such a controlling concept.

We would first like to mention that we are not necessarily trying to show that PWM is superior to AM under all circumstances. We rather want to show that PWM enables behaviours that researches

in the field of synthetic biology have strived to achieve using synthetic gene circuits, for example the “linearizer” circuit mentioned by Reviewer 2 (Nevozhay et al. 2009). Despite many recent advances in the field, engineering of circuits with defined properties is still usually very challenging. Thus, we would argue that the ability to achieve similar behavior using simple gene expression systems can have a big impact. We note that similar optogenetic expression systems were established in a variety of organisms (Müller et al. 2015) and we are very confident that our results are portable to other systems as the mechanisms we describe are not based on VP-EL222 itself.

Regarding the co-regulation of genes for synthetic biology applications, it appears widely established that tuning of relative expression levels is crucial (for example in many metabolic engineering applications (Du et al. 2012)). Furthermore, dynamic regulation of gene expression in these fields is becoming more and more prevalent (Brockman & Prather 2015). PWM regulation enables us to perform such regulation while keeping expression ratios constant and using very simple promoter libraries that do not need to be specifically tuned. Of course, the degree to which such a regulation strategy can be beneficial is case dependent. Lastly, the linearity of the response to PWM makes it easier to tune expression levels in a predictive fashion.

For the importance of reducing or tuning of cell-to-cell variability in synthetic biology applications, there is still less evidence but it has been proposed that phenotypic heterogeneity impacts bioprocess performance (Delvigne et al. 2014). It was argued previously that our relative lack of understanding of the phenotypic consequences of variability at least partly stems from the difficulties in tuning variability and mean expression independently (Liu et al. 2016). PWM regulation gives us the ability to do so in an easy fashion. Again, we cannot strictly talk about superiority of PWM compared to AM. Rather, the use of both PWM and AM can give us important information about the influence of variability on a given process and optimal expression levels resulting from this mapping. For example, under improved experimental conditions, we found evidence that cell-to-cell variability in the expression of the metabolic enzyme Ura3 can be beneficial for cellular growth at very low expression levels (see below).

Lastly, we would like to note the basic research / systems biology implications of our work. To the best of our knowledge, variability reduction by pulsatile signaling has not been previously demonstrated experimentally. Our results suggest the possibility that variability reduction may be a functional role of pulsing in cellular signaling and we now discuss this briefly in the text (Discussion section). We further provide a novel mechanism for the observed noise reduction.

Taking all of these together, we would argue that coming up with another synthetic biology ‘application’ should not be strictly necessary to advance our point. Nevertheless, we have improved on the previous application of the system towards the study of the phenotypic consequences of expression variability on cell growth (see below).

2. Related to the above comment, in page 4 (line 162) the authors tried to illustrate the application of PWM for metabolic gene control. However, the data provided here is insufficient as a proof of concept: In an analysis where variability is the major factor, two repeats is far from sufficient to draw a sound conclusion. In addition, in Fig. 3h it can also be seen that there is considerable overlap between the two cases, and the separation of the two lines is only possible after taking the average. Thus, while this is an interesting topic, the current version of the manuscript lacks a convincing example to make the case.

We have now improved upon the experiments in two main ways. First, prompted by the comment of Reviewer 1, we have directly labeled URA3 with mCitrine and do thus not rely on a proxy reporter anymore. Furthermore, we have changed the VP-EL222 responsive promoter used to drive the expression of URA3. This enabled us to measure a more complete dose response curve which now more clearly shows that the dose-response curve between URA3 expression and growth differs between AM and PWM (**Fig. 5**). It further uncovers a beneficial effect of noise at low

expression levels and still shows that PWM enables maximal growth at lower expression levels. We further use a simple model to illustrate how these observations can result from sharp dose-response curves / all-or-none behaviour on the single cell level.

We performed 3 replicates for the experimental condition. We provide a plot that shows the results of all experiments as single data points in **Supplementary Fig. 13d**. The response to AM and PWM was always measured in the same experiment to exclude systematic biases. We would like to add that the experiments are performed in tube cultures with cellular populations of > 100,000 cells at the start of the experiment and thus, the variability on the single cell level should not strongly affect experimental reproducibility. Importantly, the observed differences in variability between AM and PWM are robust between experiments.

3. In the study, only a single protein with a long half-life was used to demonstrate that the in-period fluctuation is small. However, to prove the point, it would be persuasive to examine proteins with reduced stabilities. Importantly, it will be interesting to see if proteins with short half-lives, which are intrinsically noisy, continue to have reduced fluctuations. If that is the case, it would have broad applications in different settings.

We have now performed further experiment using a destabilized variant mCitrine for which we calculate a degradation rate of 30 min without accounting for dilution (Fig. 1g, Supplementary Fig. 2d). We show experimentally that there is indeed input tracking for a destabilized fluorescent protein, however the changes across a PWM period are rather low indicating that successful PWM regulation is possible with such proteins (Fig. 1h). We found that variability (measured by the CV) at lower input levels is increased when compared to the stable fluorescent protein and that variability can be attenuated by PWM also in the case of non-stable proteins (**Supplementary Fig. 12e**).

4. One of the major claims of this article is that gene expression variability can be tuned ("precisely", as stated in the abstract). This point isn't sufficiently shown by the results though. What is shown by the data (Fig. 3a, e) is that different periods of PWM result in different variability in a nonlinear way, which doesn't give much flexibility in achieving a specific variability by tuning the period in a quantitative way.

Indeed, we have not explored this precise regulation experimentally and have thus removed this statement ("precisely") from the abstract as we do not see the model-predictive regulation of variability as our major contribution. We have however now extended our modeling efforts and identified a simple stochastic model that can quantitatively describe the observed expression variability (**Fig. 4** and **Supplementary Fig. 12**, see also below at comment 5). We are thus highly confident that such precise regulation is indeed possible and state the following in the main text:

"We found that the model further nicely predicted the expression variability under different PWM periods (**Fig. 4e**), PWM amplitudes (**Supplementary Fig. 12e**), and for different reporter protein degradation rates (**Supplementary Fig. 12e**), which may open the way to a model-predictive regulation of expression mean and variability."

5. As for the more fundamental exploration of the mechanism, the reduction of variability by PWM is analyzed compared to AM, but the mechanism about how variability can be tuned by PWM, namely, how changing the period can lead to changes in variability, is lacking. There is thus a mismatch between the major claim, and the main efforts in the analysis provided. I would suggest the authors to examine how CV correlates with PWM period in a quantitative way with simulations. It will also be valuable to provide a mechanism to explain why a longer period results in a lower variability. Additionally, a more "precise" modeling approach (stochastic model) can be a more theoretically sound framework to study the variability for the experimental systems.

Our analysis suggests that the ability to tune variability using the PWM period stems from active VP-EL222 concentrations residing at intermediate levels for extended time intervals at low PWM

periods and/or short duty cycles. We do now explicitly state this in the text and we have added a new panel F to Figure 3 that aims to illustrate this point by showing the temporal changes in TF activity, transcription rate, and transcription variability for different PWM periods based on simulations.

We have now also performed simulations using a stochastic model that both incorporates intrinsic and extrinsic / VP-EL222 variability. VP-EL222 variability is modeled by explicitly accounting for its expression dynamics using a simple stochastic gene expression model, meaning that we model the extrinsic noise for the reporter gene as intrinsic noise upstream in the cascade. We found that doing so can nicely recapitulate / predict extrinsic noise in a quantitative manner. We further show that the intrinsic variability can be nicely recapitulated using a standard two-state promoter model. Importantly, this model also quantitatively recapitulates / predicts the effect of PWM period on expression variability under different conditions (**Fig. 4** and **Supplementary Fig. 12**).

Minor points:

1. Fig. 1e: it seems that the rate of mRNA production is incorrect.

We have corrected this mistake.

References:

- Bodvard, K. et al., 2011. Continuous light exposure causes cumulative stress that affects the localization oscillation dynamics of the transcription factor Msn2p. *Biochimica et Biophysica Acta (BBA) - Molecular Cell Research*, 1813(2), pp.358–366.
- Brockman, I.M. & Prather, K.L.J., 2015. Dynamic metabolic engineering: New strategies for developing responsive cell factories. *Biotechnology journal*, 10(9), pp.1360–1369.
- Delvigne, F. et al., 2014. Metabolic variability in bioprocessing: implications of microbial phenotypic heterogeneity. *Trends in biotechnology*, 32(12), pp.608–616.
- Du, J. et al., 2012. Customized optimization of metabolic pathways by combinatorial transcriptional engineering. *Nucleic acids research*, 40(18), p.e142.
- Hansen, A.S. & O’Shea, E.K., 2013. Promoter decoding of transcription factor dynamics involves a trade-off between noise and control of gene expression. *Molecular systems biology*, 9, p.704.
- Liu, J., François, J.-M. & Capp, J.-P., 2016. Use of noise in gene expression as an experimental parameter to test phenotypic effects. *Yeast*, 33(6), pp.209–216.
- Müller, K. et al., 2015. Optogenetics for gene expression in mammalian cells. *Biological chemistry*, 396(2). Available at: <http://dx.doi.org/10.1515/hsz-2014-0199>.
- Nevozhay, D. et al., 2009. Negative autoregulation linearizes the dose-response and suppresses the heterogeneity of gene expression. *Proceedings of the National Academy of Sciences of the United States of America*, 106(13), pp.5123–5128.

REVIEWERS' COMMENTS:

Reviewer #1 (Remarks to the Author):

Comments to the authors:

The authors have addressed the major concerns from my first review, namely:

1) They have improved the URA3 regulation experiments by directly tagging URA3 with mCitrine so that URA3 concentration can be directly related to cell growth. Furthermore, they repeated these experiments with a promoter that allows for a wider dose-response curve so that the difference between AM and PWM control of URA3 expression is more evident.

2) They have justified their use of the budded cell population by confirming that these cells seem to have a higher degree of plasmid retention (using colony counts in plating assays on media with and without leucine).

3) They have responded to both my and reviewer #2's concerns regarding how growth rates were calculated and the effect of phototoxicity on growth rate. These improvements are indicated in Supplemental Figure 1 and Supplemental Figure 13.

Additionally, the use of a stochastic model to explore the causes of extrinsic and intrinsic variability is a good addition to this study. While the suggestion that pulsatile TF activity reduces variability from transcriptional bursting requires further study, it is an intriguing avenue for future research and doesn't need to be ironed out for this study. The discussion has been improved to contain a more nuanced and complete discussion of the results in the context of recent studies, including studies using optogenetics for dynamic metabolic regulation and the seminal studies from the O'Shea lab on the role of TF pulsing in gene expression.

Minor Comments:

- Please check the caption for Figure 1g. "Ideally, PWM should not result in significant"....what?
- Show the mCitrine data for experiments analogous to the mKate experiments in Supp Fig 2
- Why is the fit of the model to the data in Supp Fig 5B so poor? I couldn't find where this figure was referenced in the text or supplemental.

Reviewer #2 (Remarks to the Author):

I would like to thank the Authors for addressing my comments satisfactorily. I would like to recommend the acceptance of this important manuscript for publication.

Reviewer #3 (Remarks to the Author):

The authors have done a great job in incorporating my previous comments, conducting supplementary experiments, and revising the manuscript accordingly. I thus recommend publication of the paper.

We would like to thank all reviewers again for the constructive and in-depth review process. We think that the manuscript has strongly benefited from the provided comments and suggestions.

In response to reviewer comments and journal requirements, we made the following small changes to the manuscript:

- We have added a new Supplementary Figure (S4) that shows time-course and dose-response data for the destabilized mCitrine variant.
- We added a short comment to Supplementary Note 1 (Section: 'Refitting of promoter-specific model parameters') and added a reference to Supplementary Figure 5 (now Supplementary Fig. 6) to the main text (see below).
- We have slightly shortened the captions of Fig. 1 and Fig.3.
- We have reformatted the numbering of the supplementary notes.

REVIEWERS' COMMENTS:

Reviewer #1 (Remarks to the Author):

Comments to the authors:

The authors have addressed the major concerns from my first review, namely:

1) They have improved the URA3 regulation experiments by directly tagging URA3 with mCitrine so that URA3 concentration can be directly related to cell growth. Furthermore, they repeated these experiments with a promoter that allows for a wider dose-response curve so that the difference between AM and PWM control of URA3 expression is more evident.

2) They have justified their use of the budded cell population by confirming that these cells seem to have a higher degree of plasmid retention (using colony counts in plating assays on media with and without leucine).

3) They have responded to both my and reviewer #2's concerns regarding how growth rates were calculated and the effect of phototoxicity on growth rate. These improvements are indicated in Supplemental Figure 1 and Supplemental Figure 13. Additionally, the use of a stochastic model to explore the causes of extrinsic and intrinsic variability is a good addition to this study. While the suggestion that pulsatile TF activity reduces variability from transcriptional bursting requires further study, it is an intriguing avenue for future research and doesn't need to be ironed out for this study. The discussion has been improved to contain a more nuanced and complete discussion of the results in the context of recent studies, including studies using optogenetics for dynamic metabolic regulation and the seminal studies from the O'Shea lab on the role of TF pulsing in gene expression.

Minor Comments:

- Please check the caption for Figure 1g. "Ideally, PWM should not result in significant"....what?

During the course of shortening the Figure captions, this sentence was now removed completely.

- Show the mCitrine data for experiments analogous to the mKate experiments in Supp Fig 2

We have added a new Supplementary Figure (S4) that shows time-course and dose-response data for the destabilized mCitrine variant. Data showing the equivalence of (stable) mCitrine and mKate2 in context of the dual-color reporter assay can be found in Supplementary Fig. 10.

- Why is the fit of the model to the data in Supp Fig 5B so poor? I couldn't find where this figure was referenced in the text or supplemental.

We are currently not completely sure about the source of the relatively poor fit. One possible source is a different response between the two promoters to short (and potentially low amplitude) TF pulses that cannot be captured by our simple gene expression model. For example, in (Hansen and O'Shea 2013) a model with three promoter states was required to recapitulate the response of different promoters to dynamic and constant inputs. We comment on this possibility in the main text, when we write:

"Another possible explanation for this behaviour may be promoter-dependent differences in the expression response to short TF pulses, which have been observed for natural promoters (Hansen and O'Shea 2013) (see also **Supplementary Note 1** 'Refitting of promoter-specific model parameters')

We now further added a comment on this topic to Supplementary Note 1 (Section: 'Refitting of promoter-specific model parameters') which is referred to after the sentence shown above. It reads:

"We found that the fit to the PWM experiment with a 7.5 min period was worse for the 2xBS-CYC180 than for the 5xBS-CYC180 promoter (**Supplementary Fig. 6**). In contrast, the response to PWM with a 30 min period was predicted well by the model for both promoters (**Fig. 2b**, **Supplementary Fig. 7**). These results could indicate promoter-dependent differences in the expression response to short (and possibly low amplitude) TF pulses that cannot be recapitulated by the simple gene expression model. Such differences have been previously observed for natural promoters (Hansen and O'Shea 2013)."

We would like to note that we have purposefully chosen a very simple gene expression model and that this model can very well recapitulate (and to a certain degree predict) most experimentally observed behaviours in a quantitative fashion.

Supplementary Fig. 5 (now S6) was cited in Supplementary Note 1 (Section: 'Refitting of promoter-specific model parameters') and is now also cited in the main text:

"However, when we analyzed the response of two promoters differing in the number of EL222 binding sites to AM, we found that they show different dose-response behaviors (**Fig. 2b**, see **Supplementary Fig. 6** for modeling results regarding the CYC180 promoter with two binding sites)."

Reviewer #2 (Remarks to the Author):

I would like to thank the Authors for addressing my comments satisfactorily. I would like to recommend the acceptance of this important manuscript for publication.

Reviewer #3 (Remarks to the Author):

The authors have done a great job in incorporating my previous comments, conducting supplementary experiments, and revising the manuscript accordingly. I thus recommend publication of the paper.

References:

Hansen, Anders S., and Erin K. O'Shea. 2013. "Promoter Decoding of Transcription Factor Dynamics Involves a Trade-off between Noise and Control of Gene Expression." *Molecular Systems Biology* 9 (November): 704.